# Mechanically induced pyroptosis enhances cardiosphere oxidative stress resistance and metabolism for myocardial infarction therapy

Yingwei Wang [1], Qi Li [1], Jupeng Zhao[1], Jiamin Chen[1], Dongxue Wu[2], Youling Zheng[1], Jiaxin Wu[1], Jie Liu[1], Jianlong Lu[1], Jianhua Zhang[2] ✉ & Zheng Wu [1] ✉

Current approaches in myocardial infarction treatment are limited by low cellular oxidative stress resistance, reducing the long-term survival of therapeutic cells. Here we develop a liquid-crystal substrate with unique surface properties and mechanical responsiveness to produce size-controllable cardiospheres that undergo pyroptosis to improve cellular bioactivities and resistance to oxidative stress. We perform RNA sequencing and study cell metabolism to reveal increased metabolic levels and improved mitochondrial function in the preconditioned cardiospheres. We test therapeutic outcomes in a rat model of myocardial infarction to show that cardiospheres improve long-term cardiac function, promote angiogenesis and reduce cardiac remodeling during the 3-month observation. Overall, this study presents a promising and effective system for preparing a large quantity of functional cardiospheres, showcasing potential for clinical application.

The 30-day mortality following myocardial infarction (MI) was 13.6% on average[1]. When MI occurs, myocardial ischemia causes a series of irreversible pathological processes, such as severe inflammation, massive cell death, and cardiac fibrosis, which ultimately lead to heart failure[2,3]. To date, many clinical and animal studies have shown cell-based therapies as promising approaches to reverse or slow MI disease progression[4–7].

To pursue satisfactory therapeutic outcomes, many effective cell processing methods have been extensively developed to improve cell bioactivities. Hanging drops, spinner flasks, and three-dimensional (3D) bioprinting have been used to prepare spheroids or organoids[8–10]. By providing mechanical cues, extracellular matrix (ECM), and soluble factors in native niches, the 3D spheroids could promote pluripotency marker expression (*Nanog*, *Oct4*, *Sox2*), cardiac lineage differentiation, paracrine secretion, anti-inflammation, and antisenescence[11–14]. To

further improve cell survival in hostile environments and their therapeutic potential, many researchers have suggested that simulating the inflammatory environment with preconditioning strategies could enhance cell resistance to adverse effects. Hypoxia and low-concentration inflammatory factor treatment are widely used preconditioning strategies[15,16]. Following preconditioning treatment, the phenotype of pretreated cells shifted in therapeutically desirable directions, and their abilities to resist inflammation were greatly enhanced[17–19]. These studies demonstrated the feasibility and effectiveness of preconditioning treatment, and they highlighted the significance of enhancing cellular bioactivities and inflammation resistance in damaged tissue regeneration.

Cardiosphere-derived cells (CDCs) are of endogenous cardiac origin[20] and possess the ability to form 3D spherical clones, cardiospheres (CSps), in vitro[21]. Compared to monolayer CDCs, CSps

[1]Key Laboratory for Regenerative Medicine, Ministry of Education, Department of Developmental and Regenerative Biology, Jinan University, Guangzhou, China. [2]Department of Cardiology, First Affiliated Hospital of Jinan University, Guangzhou, China. ✉e-mail: Zhangjh_jnu@126.com; wuzheng@jnu.edu.cn

possessed improved growth factor secretion and cardiac regeneration potential, making them a good cell source for MI therapy[22,23]. Previous reports demonstrated that CSps could regulate the inflammation of infarcted myocardium through immunomodulatory effects[24–26], and it was proposed that the CDCs polarized macrophages away from the M1 phenotype but not toward a classical M2 state, but to a distinct cardioprotective phenotype that promotes the survival of ischemic cardiomyocytes[27]. Also, the secretion of various growth factors and bioactive molecules from CSps could involve in the vessel network rebuilding process, which were beneficial for reducing ventricular adverse remodeling and hypertrophy[28–31]. These characteristics make CSps a good cell source for MI therapy. Previously, preconditioning CSps with pericardial fluid obtained from myocardial infarction was prepared by our colleagues Zhang et al., and the paracrine function and survival rate of the pericardial fluid-pretreated CSps dramatically increased, exhibiting significant improvement of MI cardiac function, and the DiR-labeled CSps showed cTnT-positive in the infarcted area, indicating the direct differentiation of CSps into cardiomyocytes[32]. Moreover, Zhang et al. reported that pericardial application could serve as a new and effective route for CSps transplantation, and this therapeutic strategy also showed favorable potential for further clinical application[33].

Cellular physiological activities could be manipulated by the properties of the contacted substrate[34–36]. Liquid-crystal patterns could directly introduce cells into a 3D environment and form cell structures in situ[37], which may be beneficial for mass spheroid production during large-scale clinical applications. In addition, the alignment of monolayer support cells could form a nematic liquid crystal pattern to induce cell death at the stress localization[38]. Pyroptosis is an inflammation-related cell death program, and the pyroptotic cells could release complex inflammatory signals to affect surrounding cells[39]. Considering the inflammatory signals released from the injured cells would be more efficient in improving the cytoprotective function of the target cells than the artificial stimulus inducer[15], the phenomenon of liquid crystal pattern to induce cell death might be applied to develop a local dynamic inflammatory milieu and turn this theoretical model into cell product preparation methods. Therefore, using a liquid crystal substrate for pretreated CSps might be a stable, convenient, and effective strategy to achieve mass production and cell function improvement.

In this work, a comprehensive optimized 3D culture platform for effective CSps production and preconditioning are developed using a new kind of cholesteric liquid crystal substrate, octyl hydroxypropyl cellulose ester (OPC)[40]. The OPC substrate could promote 3D spheroid formation and induce cell death with its unique properties of liquid crystals, and the internal cells in the spheroid could be activated by inflammatory factors secreted from the external cells. The bioactivities, metabolism, and function of OPC-CSps are analyzed, and their therapeutic effects on heart function, angiogenesis, inflammatory infiltration, and ventricular remodeling are evaluated in a rat MI model.

## Results

### Liquid-crystal OPC possessed mechanical responsiveness

The synthesis schematic diagram of OPC is shown in Fig. 1a. The polarized light microscopic images revealed that the OPC displayed the characteristics of liquid crystals, including birefringence, fissures, and fingerprint-like texture (Fig. 1b). The atomic force microscope results showed that the surface of the OPC substrate was a nonflat profile with wavy bulges. The height of the grains ranged from 20–35 nm, and the roughness (root mean square height, Sq) was $2.33 \pm 0.29$ nm (Fig. 1c). The static contact angle of the OPC substrate was $106.49 \pm 2.36°$, indicating that it had a hydrophobic surface (Fig. 1d). The effect of shear force on the OPC substrate surface was examined. As the X-ray diffraction (XRD) results showed, there were two diffraction peaks before shearing, and peaks at approximately $2\theta$

of 20–22° were enhanced following the application of shear force, indicating the rearrangement of the liquid crystal unit (Fig. 1e). In addition, the average crystallization rate and the grain size of the vertical (002) crystal plane were calculated according to the XRD results, and the average crystallization rate increased from 17.91% to 21.48%, and the grain size of the vertical (002) crystal plane increased from 0.69 nm to 1.14 nm. The viscoelasticity of the OPC substrate was examined by a stress-controlled rheometer, and the phase transition was observed. At 1–10 rad, the loss modulus (G″) was higher than the energy storage modulus (G′), and OPC preferred viscous deformation behavior. In contrast, at 10–100 rad, the energy storage modulus (G′) was higher than the loss modulus (G″), indicating an elastic deformation tendency (Fig. 1f). The OPC substrate was nontoxic in cell culture (Fig. 1g).

### The OPC substrate promoted CSps formation and progenitor phenotypes

The formation of CSps on the polystyrene (PS) substrate, the ultralow attachment (ULA) substrate and the OPC substrate was observed, and different shapes of CSps were acquired (Fig. 2a and Supplementary Fig. 1). The adherent CDCs on the PS substrate converged and formed regular spherical aggregates, and supporting cells surrounded at the base of the PS-CSps. The ULA-CSps were formed by suspended single-cell stacking, and they developed into noncircular, oval, and irregular shapes. On the OPC substrate, CDCs initially attached to the substrate and gradually aggregated to form regular and circular spheroids, and no supporting cells were observed around the OPC-CSps. During the formation of CSps, the sizes of OPC-CSps showed a stable growth tendency compared with the obvious increase of the PS-CSps and the ULA-CSps (Fig. 2b and Supplementary Fig. 1). Following a 3-day cultivation, the sizes of PS-CSps, ULA-CSps, and OPC-CSps were $280.96 \pm 40.56$ μm, $203.34 \pm 69.36$ μm, and $120.29 \pm 15.34$ μm, respectively. In addition, the spheroid density on the OPC substrate was significantly higher than that on the other two substrates (Fig. 2c).

The expression levels of CSps surface markers were analyzed. Compared to the PS group (KDR:$3.15 \pm 0.85\%$, Sca-1: $12.61 \pm 3.63\%$) and the ULA group (KDR: $3.76\% \pm 0.79\%$, Sca-1: $11.13 \pm 2.39\%$), the OPC group exhibited the highest expression of KDR ($8.55 \pm 1.20\%$) and Sca-1($26.17 \pm 4.17\%$) ($P < 0.05$). In addition, the ULA group (CD31:$11.74 \pm 0.89\%$, CD34: $13.91 \pm 1.64\%$) and the OPC group (CD31: $11.99 \pm 1.07\%$, CD34: $14.47 \pm 0.85\%$) showed an increase in the expression of CD31 and CD34 when compared to the PS group (CD31: $2.68 \pm 1.24\%$, CD34: $1.80 \pm 0.55\%$) ($P < 0.05$), along with a decrease in CD90 (PS: $98.47 \pm 0.65\%$, ULA: $68.37 \pm 4.81\%$, OPC: $6.87 \pm 1.01\%$) ($P < 0.05$) and no significant difference in CD105 compared to the PS group (PS: $45.73 \pm 7.03\%$, ULA: $37.03 \pm 3.50\%$, OPC: $39.53 \pm 8.51\%$) (Fig. 2d).

### OPC-induced pyroptosis improved CSps cellular bioactivities and paracrine effects

The expression of caspase-1 was observed in the peripheral cells of the OPC-CSps, while no positive signals were observed in the PS group and the ULA group (Fig. 3a). The transcription levels and protein expression levels of the pyroptosis-related key factors caspase-1 and IL-1β in the OPC group were both significantly upregulated compared with those in the PS group and the ULA group ($P < 0.05$) (Fig. 3b–d). The transmission electron microscope (TEM) analysis was performed to evaluate the cellular ultrastructure. Highly cell aggregation was observed in the ULA-CSps, while there was certain cell-cell space remained in the center of the OPC-CSps (Supplementary Fig. 2). Normal structures of the nucleus, mitochondria, and rough endoplasmic reticulum were observed in the PS group. However, endoplasmic reticulum dilatation, degranulation, and swollen mitochondria were observed in the cells from the ULA group. Moreover, cells in the OPC-CSps showed normal nuclear structures and abundant normal mitochondria. In contrast to the PS group and the ULA group, many

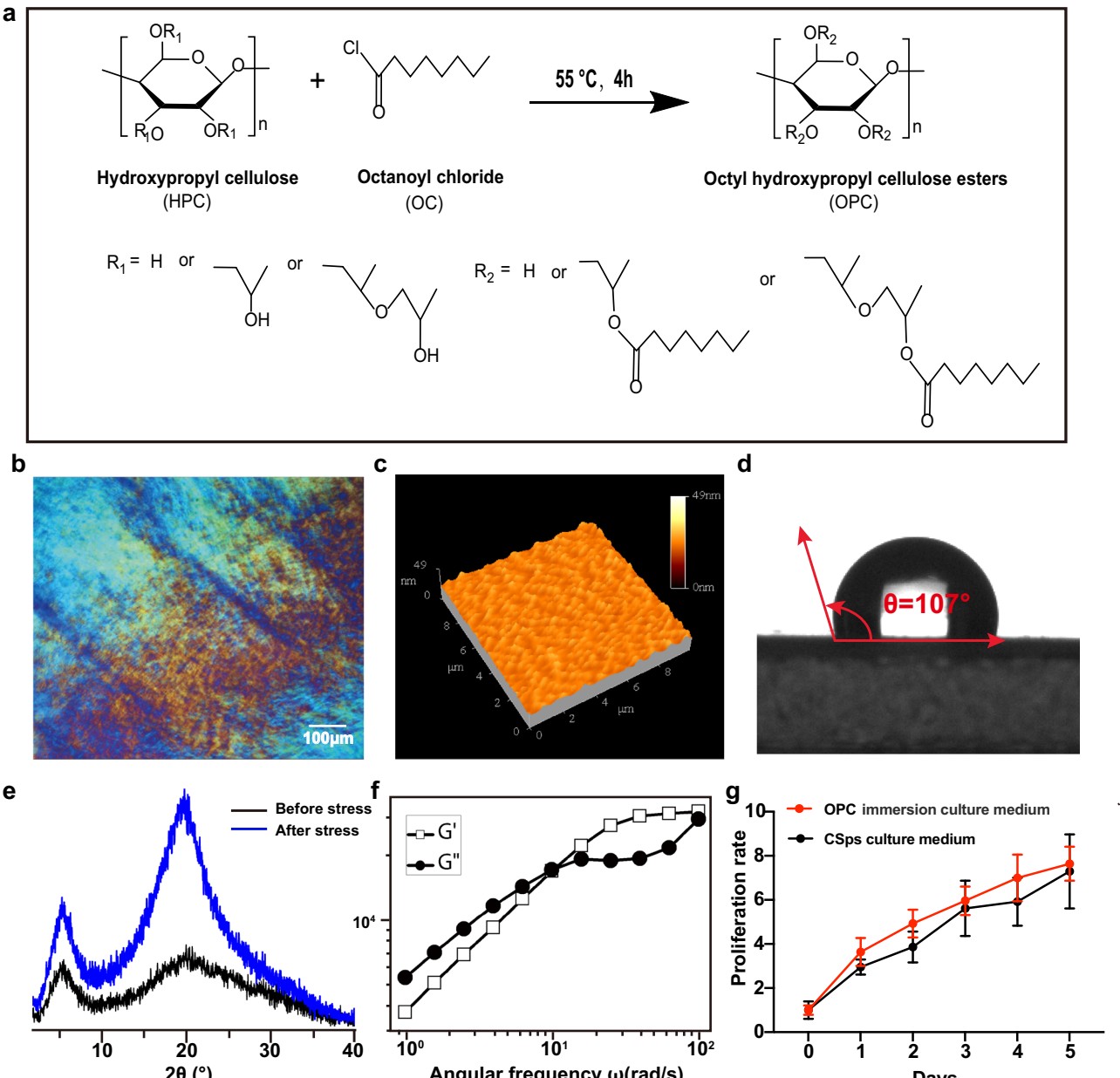

**Fig. 1 | OPC showed liquid crystal characteristics and mechanical responsiveness. a** Schematic diagram of OPC synthesis. **b** Representative image of OPC under polarized light microscopy, and 10 independent samples were observed. **c** Representative atomic force microscopy results of OPC ($n = 5$). **d** The static contact angle ($\theta$) of OPC ($n = 7$). **e** The XRD results of OPC before and after shear force application. **f** The storage modulus (G′) and loss modulus(G″) over 1–100 angular frequency measured by a stress-controlled rheometer. **g** The results of the OPC toxicology test ($n = 5$ biologically independent samples). All data are shown as the mean ± SD, two-way ANOVA (**g**).

microvesicles could be observed on the membrane surface (Fig. 3e). Following a 3-day cultivation, the cell survival rate of the ULA group was markedly lower than that of the PS group. The cell survival rate of the OPC group was significantly higher than that of the ULA group ($P < 0.05$), and it showed no significant difference from the PS group (Fig. 3f). The expression of Ki67 was observed in the center of the PS-CSps and the OPC-CSps, while the Ki67+ cells were mostly found in the periphery of the ULA-CSps (Supplementary Fig. 3). Additionally, the proliferation ability of the CDCs from OPC-CSps was significantly higher in the OPC-CSps than that from ULA-CSps ($P < 0.05$), and it showed no significant difference from the PS-CSps (Fig. 3g). In addition, the transcription levels of the cell pluripotency markers *Oct4*, *Nanog*, and *Sox2* (Fig. 3h) and the paracrine-related genes *VEGF*, *HGF*,

*IGF-1*, and *bFGF* dramatically increased following OPC culture compared with the PS and ULA groups ($P < 0.05$) (Fig. 3i).

**OPC substrate improved the mitochondrial function of CSps by enhancing oxidative phosphorylation and decreasing glycolysis**
The results of RNA-sequencing (RNA-Seq) analysis showed that there were 1251 differentially expressed genes (DEGs) between the ULA group and the OPC group (Fig. 4a), and 232 DEGs were not among the DEGs of PS vs. ULA or PS vs. OPC (Fig. 4b). Kyoto Encyclopedia of Genes and Genomes (KEGG) enrichment analysis showed that metabolic pathways, glycolytic/glycogenic pathway, and the HIF-1 signaling pathway are the top 3 significant signaling pathways of the DEGs between the OPC-CSps and the ULA-CSps (Fig. 4c). Moreover, gene set

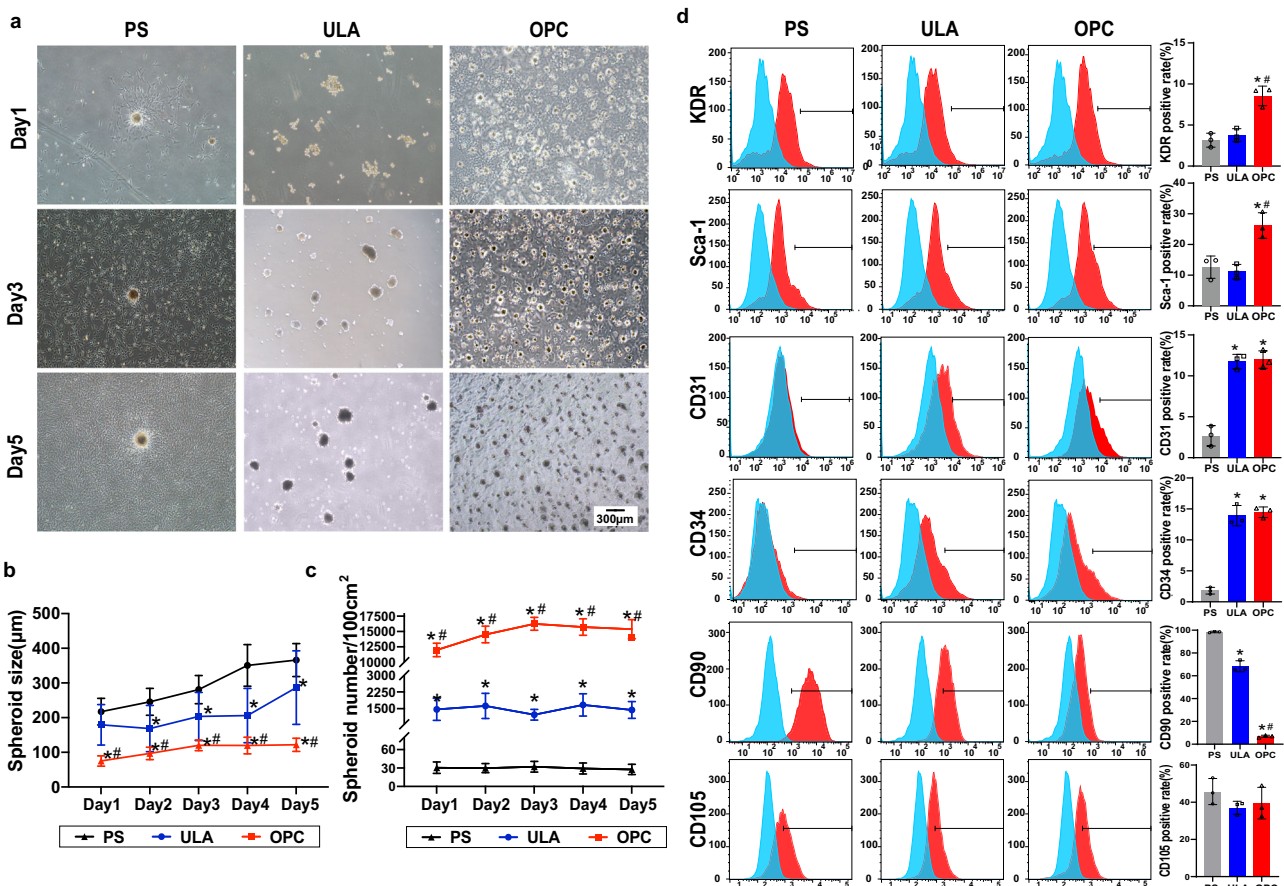

**Fig. 2 | OPC-CSps displayed controllable spheroid size and favorable progenitor cell phenotypes. a** Morphological change in CSps on the PS, ULA, and OPC substrates. **b** Quantification results of the spheroid size in 5-day cultivation on the PS, ULA, and OPC substrates (*n* = 30 images from 6 independent experiments). **c** Quantification results of the CSps density in 5-day cultivation on each substrate

(*n* = 8 from 4 independent experiments). **d** Phenotype characterization of CSps from each group after 3 days of cultivation. The proportions of positive cells relative to the isotype control are shown (*n* = 3 biologically independent samples). All data are shown as the mean ± SD, *P < 0.05 vs. PS, #P < 0.05 vs. ULA. One-way ANOVA (**d**) or two-way ANOVA (**b**, **c**).

enrichment analysis (GSEA) also revealed the downregulation of the hypoxic and glycolytic components in the OPC-CSps compared to the ULA-CSps (Fig. 4d). The oxidative phosphorylation genes in the OPC group, including *CS*, *COXII*, *IDH2*, *SDHA*, and *MDH2*, were notably upregulated compared with those in the PS and ULA groups (*P* < 0.05). Compared to the ULA group, the key genes of the glycolytic pathway, *HK2*, *LDHA*, and *PFKL*, dramatically decreased in the OPC group (*P* < 0.05) (Fig. 4e). In addition, the transcription levels of these three genes showed no difference between the PS group and the OPC group. Compared to the PS-CSps and the ULA-CSps, the 2-(N-(7-nitrobenz-2-oxa-1,3-diazol-4-yl) amino)−2-deoxyglucose (2-NBDG) uptake level of the OPC-CSps significantly decreased (*P* < 0.05) (Fig. 4f). The lactate production by the OPC-CSps was markedly lower than that by the ULA-CSps, and it was significantly higher than that by the PS-CSps (*P* < 0.05) (Fig. 4g). Among these three groups, the OPC-CSps had the highest ATP production level (Fig. 4h).

The TEM results showed that the mitochondria in the CDCs of ULA-CSps exhibited obvious swelling, vacuolization, and cristae breakage, while the mitochondria in the CDCs of OPC-CSps and the PS-CSps maintained normal cristae morphology. Furthermore, the density of mitochondria was significantly increased in the OPC group compared to the PS group and the ULA group (*P* < 0.05) (Fig. 4i). Mitochondrial membrane potential levels of the CDCs in CSps were evaluated by immunofluorescence staining of rhodamine 123 fluorescence intensity, and the membrane potential level was significantly

enhanced in the cells of OPC group compared to those of the PS group and the ULA group (*P* < 0.05) (Fig. 4j).

## OPC-CSps showed enhanced oxidative stress resistance and anti-inflammatory effect

$H_2O_2$ stimulation was used to test cellular oxidative stress resistance. As shown in Fig. 5a, b, after exposure to $H_2O_2$ for 24 h, the fluorescence intensity of the ULA group was significantly lower than that of the PS group(*P* < 0.05). Moreover, the fluorescence intensity of the OPC group further decreased compared to the PS group and the ULA group, suggesting that OPC-CSps generated the least reactive oxygen species (ROS) and superoxide under an oxidative stress environment. Following 12 h of $H_2O_2$ stimulation, the percentage of viable cells in the OPC group (73.7 ± 1.4%) was significantly higher than that in the PS group (59.7 ± 7.5%) and the ULA group (59.2 ± 5.4%) (*P* < 0.05) (Fig. 5c). There was no difference in the cell survival rate between the PS-CSps and the ULA-CSps. Following 24 h of $H_2O_2$ stimulation, a similar trend in the survival rate was observed among the three groups, and the survival rates of the PS, ULA, and OPC groups were 1.5 ± 0.7%, 5.4 ± 2.1%, and 31.7 ± 4.7%, respectively (Fig. 5d). The anti-inflammatory effects were also examined. Following 12 h of TNF-α stimulation, the survival rates of the PS, ULA, and OPC groups were 50.0 ± 3.7%, 53.8 ± 4.5%, and 66.3 ± 3.0%, which were significantly lower than the control group, while the cells from the OPC-CSps group exhibited the best anti-inflammatory effect among three groups (Fig. 5e).

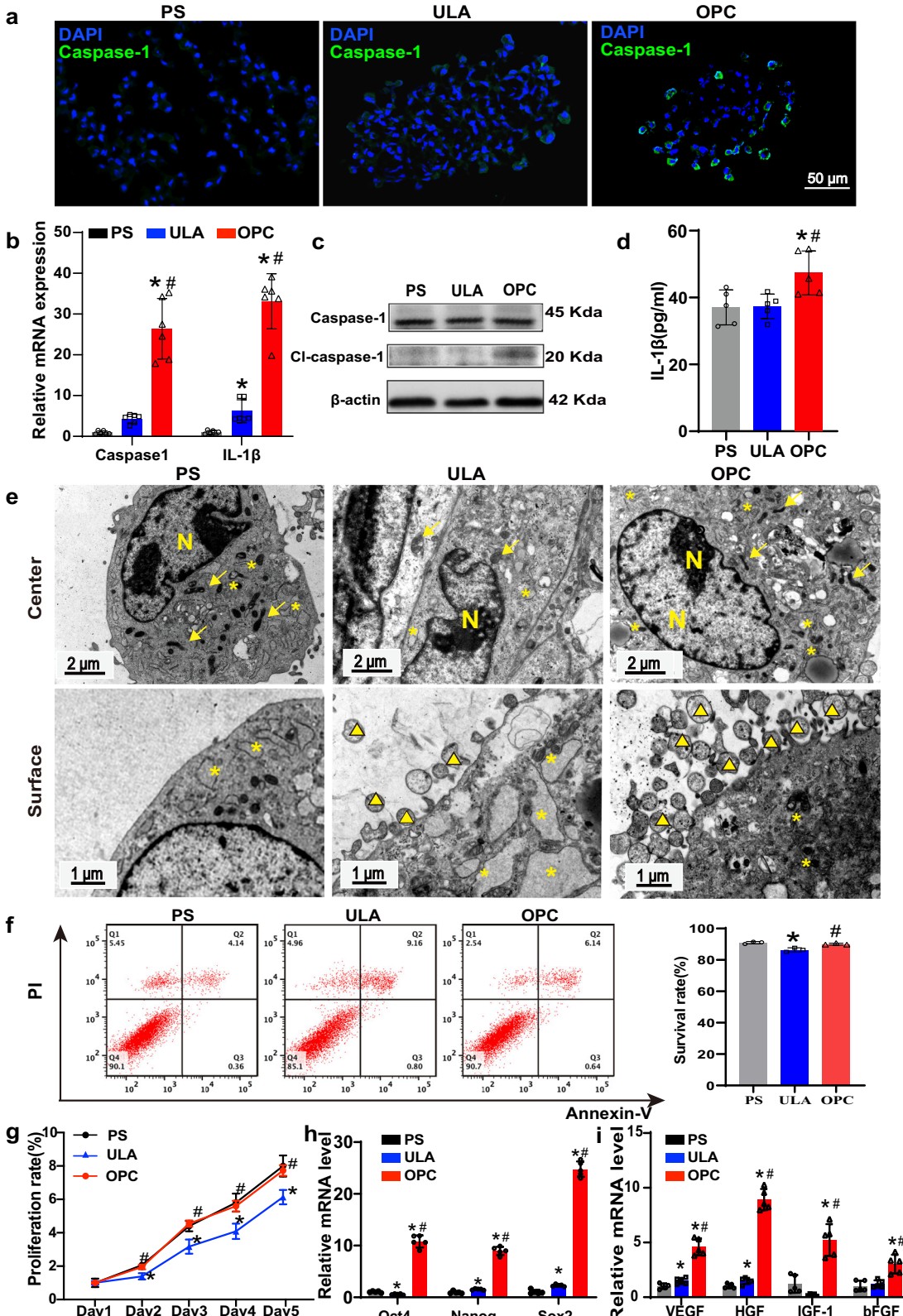

## Long-term cardiac function improvement following OPC-CSps transplantation in a rat MI model

In vivo live imaging was performed to detect the survival rate of transplanted OPC-CSps within the infarct area. The survival rate of transplanted OPC-CSps was 53.83 ± 9.01% at week 2 and 16.18 ± 3.68% at week 4 (Fig. 6a, b). According to the echocardiography results, serious motor dysfunction of the anterior ventricular wall was observed in the vehicle group following ligation. However, motor function was maintained in the OPC-CSps group compared to the vehicle group. The results also revealed that the OPC-CSps group significantly improved cardiac function starting at week 4, and this tendency was sustained throughout the 12-week observation (Fig. 6c). Compared to the vehicle group, the OPC-CSps group showed a remarkable increase in left ventricular fractional shortening (LVFS) and

**Fig. 3 | OPC-induced pyroptosis improved CSps cellular bioactivities and paracrine effects. a** Representative image of caspase-1 immunofluorescence staining results of each group. **b** The mRNA transcription levels of caspase-1 and IL-1β (*n* = 5 biologically independent samples). **c** The protein expression levels of caspase-1 and cl-caspase-1 tested by western blot. **d** The concentration of IL-1β in the cell supernatant after 3 days of cultivation of each group (*n* = 5 biologically independent samples). **e** Representative TEM images of the cell ultrastructure in CSps, and 3 biologically independent samples were observed. N: nucleus, yellow arrows indicate mitochondria, yellow asterisks (*) indicate endoplasmic reticulum,

and yellow triangles (△) indicate microvesicles. **f** Annexin/PI analysis results of the cell survival rate (*n* = 3 biologically independent samples). **g** Proliferation assay of the CDCs isolated from the PS-CSps, ULA-CSps, and OPC-CSps (*n* = 5 biologically independent samples). **h** The mRNA transcription levels of *Oct4*, *Nanog*, and *Sox2* (*n* = 5 biologically independent samples). **i** The mRNA transcription levels of *VEGF*, *bFGF*, *HGF*, and *IGF-1* (*n* = 5 biologically independent samples). All data are shown as the mean ± SD, *$P < 0.05$ vs. PS, #$P < 0.05$ vs. ULA, one-way ANOVA (**d**, **f**) or two-way ANOVA (**b**, **g**, **h**, **i**).

left ventricular ejection fraction (LVEF) ($P < 0.05$) (Fig. 6d, f). The LVEF and LVFS at week 12 relative to week 4 of the vehicle group decreased by 2.74 ± 2.33% and 2.51 ± 1.24%, respectively. In contrast, the OPC-CSps group showed a 10.12 ± 2.57% improvement in LVEF and 4.54 ± 2.33% in LVFS at week 12 relative to week 4 (Fig. 6e, g). In addition, compared to the vehicle group, the OPC-CSps group showed a significant reduction in left ventricular internal diameters both in systole (LVIDs) and diastole (LVIDd) at week 8 and week 12 ($P < 0.05$) (Fig. 6h–k).

### OPC-CSps protected infarcted myocardium from inflammation, apoptosis, and hypertrophy

Compared to the vehicle group, the OPC-CSps group exhibited a significant cardioprotective effect (Fig. 7a, b). Clear vascular structures at the infarct region border were observed in the vehicle group and the OPC-CSps group. However, compared to the vehicle group, there were fewer perivascular collagens in the OPC-CSps group (Fig. 7a). Compared to the sham group, an increase in the size of the infarct area (29.25 ± 3.63%) and a decrease in myocardial tissue retention (23.66 ± 3.52%) were observed in the vehicle group. Compared to the vehicle group, the OPC-CSps group showed a significant decrease in infarct area size (18.54 ± 3.19%) and an increase in retained myocardial tissue (41.69 ± 7.99%) (Fig. 7c, d). Compared to the vehicle group (0.60 ± 0.06 mm), the thickness of the left ventricular wall was higher in the OPC-CSps group (1.27 ± 0.19 mm), which reached 46% of the normal left ventricle wall thickness (2.71 ± 0.28 mm) (Fig. 7e).

For cardiac inflammation evaluation, CD68+ macrophages were calculated. In the sham group, macrophage infiltration was scarcely observed, and the number of CD68+ macrophages in the vehicle group significantly increased compared to that in the sham group ($P < 0.05$). In the OPC-CSps group, the number of CD68+ macrophages significantly decreased compared to that in the vehicle group ($P < 0.05$) (Fig. 7b, f). The structure and distribution of the vessels in the LV wall were observed by α-SMA and CD31 immunofluorescence co-staining. The vessel lumen could be obviously observed in the sham, vehicle, and OPC-CSps groups. Compared to the sham group, the vessel density of the vehicle group and the OPC-CSps group both significantly increased ($P < 0.05$). Compared to the vehicle group, a significantly higher vascular density ($P < 0.05$) and mature large-diameter blood vessels (>100 μm) were observed in the OPC-CSps groups (Fig. 7b, g, Supplementary Fig. 4). Terminal deoxynucleotidyl transferase-mediated dUTP nick end labeling (TUNEL) staining results showed that the percentage of apoptotic cells in the vehicle group (62.71 ± 10.17%) was significantly higher than that in the sham group, and a significant decrease was observed in the OPC-CSps group (29.11 ± 9.54%) ($P < 0.05$) (Fig. 7b, h).

As the wheat germ lectin (WGA) staining results showed, the average cardiomyocyte cross-sectional area was significantly higher in the infarct zone, border zone, and remote zone in the vehicle group than in the sham group ($P < 0.05$). However, in the border zone and the remote zone, the OPC-CSps group exhibited smaller cardiomyocyte sizes than the vehicle group ($P < 0.05$), and no significant differences were observed between the sham group and the OPC-CSps group in all areas measured (Fig. 7i–l).

## Discussion

Substantial progress in MI cell therapy has been widely reported, and improving transplanted cell survival and therapeutic outcomes remain the key issues to be addressed. Optimizing the culture system had great significance in obtaining abundant transplanted cells with favorable bioactivities. This study aimed to improve the therapeutic potential of CSps by optimizing their culture substrate. The prepared OPC substrate was a kind of cholesteric liquid crystal obtained by the esterification between HPC and OC. When cultured on the OPC substrate, CDCs could spontaneously form homogenous 3D spheroids at a high density, which was beneficial for quickly acquiring sufficient CSps in clinical applications. Compared with the PS substrate and the ULA substrate, CSps cultured on the OPC substrate could be activated and exhibited a superior paracrine effect, enhanced metabolic state, and improved oxidative stress resistance in the unique pyroptosis microenvironment. In a rat MI model, CSps prepared by OPCs showed a long-term cardioprotective effect within 12 weeks. A decrease in host cell apoptosis, improvement in angiogenesis, and reduction in ventricular remodeling were observed following OPC-CSps transplantation.

For producing highly functional CSps, preparing the proper cell culture substrate is the first and vital step. In this study, liquid-crystal OPC was synthesized using HPC as the rigid chain and OC as the flexible chain (Fig. 1). An optical texture formed by lattice defects was observed under the polarizing microscope, which was due to the characterization of entropy-induced phase transitions of OPC. After being subjected to an external shear force, the rigid chains of liquid crystal materials maintain molecular orderliness, while the flexible chains adjust their orientation in response to mechanical variations. The change in orientation of flexible chains not only improved the orderliness of materials but also drove the change in molecular position ordering of rigid chains when such changes accumulated to a certain extent. These alterations eventually led to the formation of lattice defects inside the material and further resulted in texture changes. The increase in the crystallinity index following shear force application implied an improvement in the orderliness of the materials, and the change in grain size was related to the change in the molecular position of the rigid chain.

In addition, the complex phase behavior of OPC is determined by the structural mosaic of rigid chains and flexible chains. Within the detection range of rheological characterization, a phase transition was observed in OPC. In the 1–10 rad region, the stress hysteresis of the flexible chain resulted in the tendency of viscous deformation. In the 10–100 rad region, the rigid chain dominated, and the stress hysteresis of the flexible chain attenuated, leading to a higher tendency of elastic deformation in OPC. The OPC substrate exhibited the highest CSps production efficiency among the three groups (Fig. 2c). The density of CSps in the OPC group was 500 times greater than that in the PS group, while it was 10 times higher than that in the ULA group. These findings showed that OPC could rearrange the liquid crystal units in response to external stress, and the characteristics of mechanical responsiveness could promote CSps formation, which could satisfy the demand for large-scale CSps production in clinical applications.

The sizes of 3D spheroids have a significant influence on cell viabilities. Spheroids within 200 μm in diameter could grow under

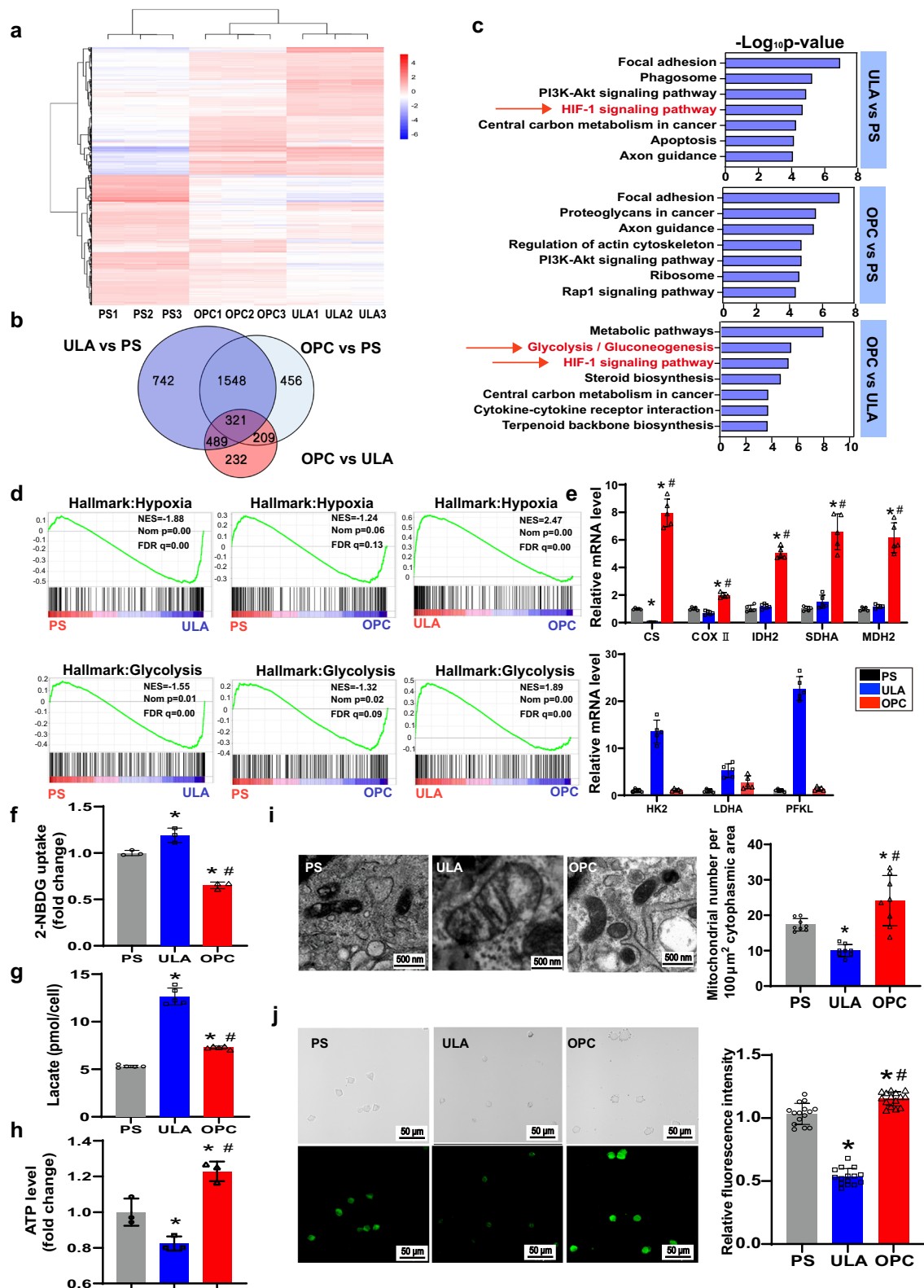

sufficient nutrition and oxygen supply, which was beneficial for delaying the formation of hypoxic cores and maintaining the viability of their internal cell populations[41,42]. In this study, different sizes of CSps were observed in the PS, ULA, and OPC groups. CSps cultured on the OPC substrate were observed to form 70–140 μm in diameter within five days (Fig. 2b), and the internal cells in the OPC-CSps maintained normal cell

ultrastructure (Fig. 3e). In contrast, the ULA spheroids were 100-470 μm in diameter and beyond the size of effective nutrient and oxygen transportation[43], so the ULA-CSps showed serious cell damage in the core of ULA-CSps with a higher portion of apoptosis (Fig. 3e, f). Owing to the mechanical cues from OPC, massive CSps with controllable size and favorable bioactivity could be obtained effectively.

**Fig. 4 | OPC substrate improved the mitochondrial function of CSps by enhancing oxidative phosphorylation and decreasing glycolysis. a** Hierarchical cluster analysis of upregulated (red) and downregulated (blue) genes after culture on different substrates for three days. **b** The DEGs from the hierarchical cluster analysis were interpreted in the Venn diagram. The DEGs with a |log2FC| ≥ 1 and a non-adjusted $P$ value ≤ 0.05 were identified by DESeq (1.28.0). **c** KEGG analysis of the top 7 significant pathways ($P$ value < 0.05). **d** GSEA revealed that the genes of the hallmark MSigDB collection were mainly enriched in hypoxia- and glycolysis-related pathways. NES, normalized enrichment score; NOM p, nominal $P$ value; FDR q, false discovery rate $q$ value. **e** mRNA transcription levels of the genes in the metabolic oxidative phosphorylation pathway (*CS, COXII, IDH2, SDHA,* and *MDH2*) and the glycolytic pathway (*HK2, LDHA,* and *PFKL*) ($n = 5$ biologically independent samples). **f** Measurement of glucose uptake by CSps using 2-NBDG ($n = 3$ biologically independent samples). **g** Lactate release level of CSps ($n = 5$ biologically independent samples). **h** The ATP levels of CDCs isolated from the CSps of each group ($n = 3$ biologically independent samples). **i** Representative TEM images of the mitochondrial morphology from each group, and the density of the mitochondria from each group was quantified ($n = 8$ images from 2 experiments). **j** Rhodamine 123 staining results of mitochondrial membrane potential, and the fluorescence intensity of the CDCs isolated from CSps of each group were measured ($n = 15$ images from 3 experiments). All data are shown as the mean ± SD, *$P < 0.05$ vs. PS, #$P < 0.05$ vs. ULA. One-sided student's $t$ test (**b**), one-sided hypergeometric test with Benjamini–Hochberg multiple testing correction (**c**), one-way ANOVA (**f**–**j**) or two-way ANOVA (**e**).

In addition, cell–cell and cell-ECM contacts also greatly affect the phenotypes of the cells in CSps. Compared to the PS group, the expression of CD31 and CD34 significantly increased in the ULA group and the OPC group (Fig. 2d), and the focal adhesion pathway of these 2 groups significantly differed from the PS group, which could be related to their higher efficiency in 3D spheroid formation. It is widely studied that the ECM components and the cell-cell contacts within the 3D cells spheroids have significant changes than the 2D culture, and these results showed CDCs in 3D spheroids may enhance their phenotypes towards endothelial cells via the change of microenvironmental cues in the ECM[44], but more studies are needed to verify this assumption. The different portions of CD90+ cells could be observed when cultured on the PS, ULA, and OPC substrates, so it is reasonable to assume that the expression level of CD90 could be regulated by the physical and chemical environments provided by each substrate. In addition, several studies have shown that a decrease in mesenchymal markers, such as CD90, could lead to an improvement in the pluripotency of spheroid cells[45,46]. In this study, compared with the PS group and the ULA group, the OPC group showed the lowest expression level of CD90 and the highest expression levels of pluripotency markers, including *Nanog, Sox2,* and *Oct4* (Fig. 3h). Furthermore, the highest expression of the cardiovascular progenitor marker KDR and the cardiac fibro-adipogenic progenitor marker Sca-1 was observed in the OPC group (Fig. 2d). In conclusion, culturing CDCs on the OPC substrate could not only obtain more progenitors in the CDC population but also facilitate their differentiation into the endothelial lineage.

In addition to favorable cell bioactivities, improving cellular resistance to oxidative stress and the inflammatory environment is another key issue in cell therapy. Following acute MI, the degradation of the extracellular matrix and the cytokines released by dead cardiomyocytes lead to serious inflammation, and massive host cells induce pyroptosis[47]. Caspase-1 and IL-1β are the classical factors of the pyroptosis signaling pathway. In this study, pyroptosis of the external cells of CSps was induced by mechanical cues from OPC, with the activation of caspase-1 and an increase in the release of IL-1β (Fig. 3a–d). Meanwhile, by receiving the proper stimulation from external pyroptosis, the internal cells with higher bioactivities exhibited an improved paracrine effect, and improved *VEGF, HGF, IGF-1,* and *bFGF* were observed (Fig. 3i). In addition, compared with the hypoxic microenvironment in CSps from the ULA group, the cellular proliferation activity of OPC-CSps was maintained (Fig. 3g). Therefore, these results demonstrated that the OPC substrate could provide proper stimulation for CSps to improve cellular paracrine effects and maintain their proliferation ability.

Cellular inflammation is directly related to oxidative stress, and improving antioxidative stress ability is vital for cell survival in MI cell therapy. When exposed to oxidative stress, the generated hydroxyl radicals can react with all biological macromolecules, causing DNA, protein, membrane damage, and ultimately cell death. Mitochondria are the center of energy metabolism, and they control many signals in cell fate programs[48]. It was reported that enhancing mitochondrial respiration and function could reduce the damage caused by oxidative stress[49]. In this study, compared to the PS group and the ULA group, the OPC-

CSps exhibited higher mitochondrial density and membrane potential levels (Fig. 4i, j). It was reported that cells in hypoxia would lead to mitochondrial damage[50], and with a compact core in the CSps, the ULA group showed lower oxidative stress resistance. In contrast, under the proper stimulation induced by OPC, the CSps in the OPC group could acquire the ability to resist oxidative stress before transplantation (Fig. 5). Taking these results together, CSps with improved oxidative stress resistance could be obtained using OPC as the culture substrate.

Furthermore, RNA-seq analysis was employed to investigate the underlying mechanism of differences in cell bioactivity and oxidative stress resistance among the three groups (Fig. 4a). The metabolic level and bioenergetic state are highly related to the availability of oxygen and nutrients[51]. In this study, DEGs between the OPC group and the ULA group were significantly enriched in the glycolysis/gluconeogenesis pathway and HIF-1 signaling pathway, but these pathways were not in the top 7 enriched pathways between the OPC group and the PS group (Fig. 4c, d). These results further proved that the structure of the OPC-CSps could satisfy the demand for internal CDC metabolism. It was reported that the enhancement of oxidative phosphorylation could surmount mitochondrial fission and functional failure[52], while glycolysis was associated with mitochondrial dysfunction[53]. In this study, the oxidative phosphorylation of OPC-CSps was enhanced, while the ULA-CSps altered their energy production toward glycolysis (Fig. 4e). Therefore, the highest ATP level was observed in the OPC group (Fig. 4h), and the OPC-CSps showed improved mitochondrial function and effective protection against cell damage in oxidative stress. In addition, the OPC group showed a significant decrease in glucose uptake levels (Fig. 4f), suggesting that the OPC-CSps may survive longer in nutrient-limited conditions. In conclusion, these results illustrated that CSps could switch toward a highly metabolically active state when cultured on the OPC substrate, which is beneficial for OPC-CSps' long-term survival in the hostile microenvironment.

With favorable cell viabilities, superior antioxidative stress, and long-term cellular survival ability, the therapeutic effect of OPC-CSps on MI was evaluated. As the results showed, the OPC-CSps group significantly improved MI cardiac function. Compared with the overtime decline tendency of the vehicle group, a consistent increase in left ventricular systolic and diastolic function was observed in the OPC-CSps group during the 12-week observation (Fig. 6). To pursue favorable outcomes in MI therapy, reducing myocardial inflammation and rebuilding the vessel network are crucial issues for protecting cardiac structure and function. In this study, owing to the effective preconditioning treatment in vitro, the OPC-CSps acquired enhanced inflammation resistance and improved regenerative function. The transplanted CSps were tracked by Dir labeling, and the results showed that 16.18% ± 3.68% of CSps survived 4 weeks following transplantation (Fig. 6a, b). Meanwhile, a significant decrease in CD68+ macrophages in the border zone was observed in the OPC-CSps group (Fig. 7b, f). Moreover, effective angiogenesis within the infarcted myocardium was widely observed at 12 weeks following MI (Fig. 7b, g). Therefore, these in vivo results demonstrated that

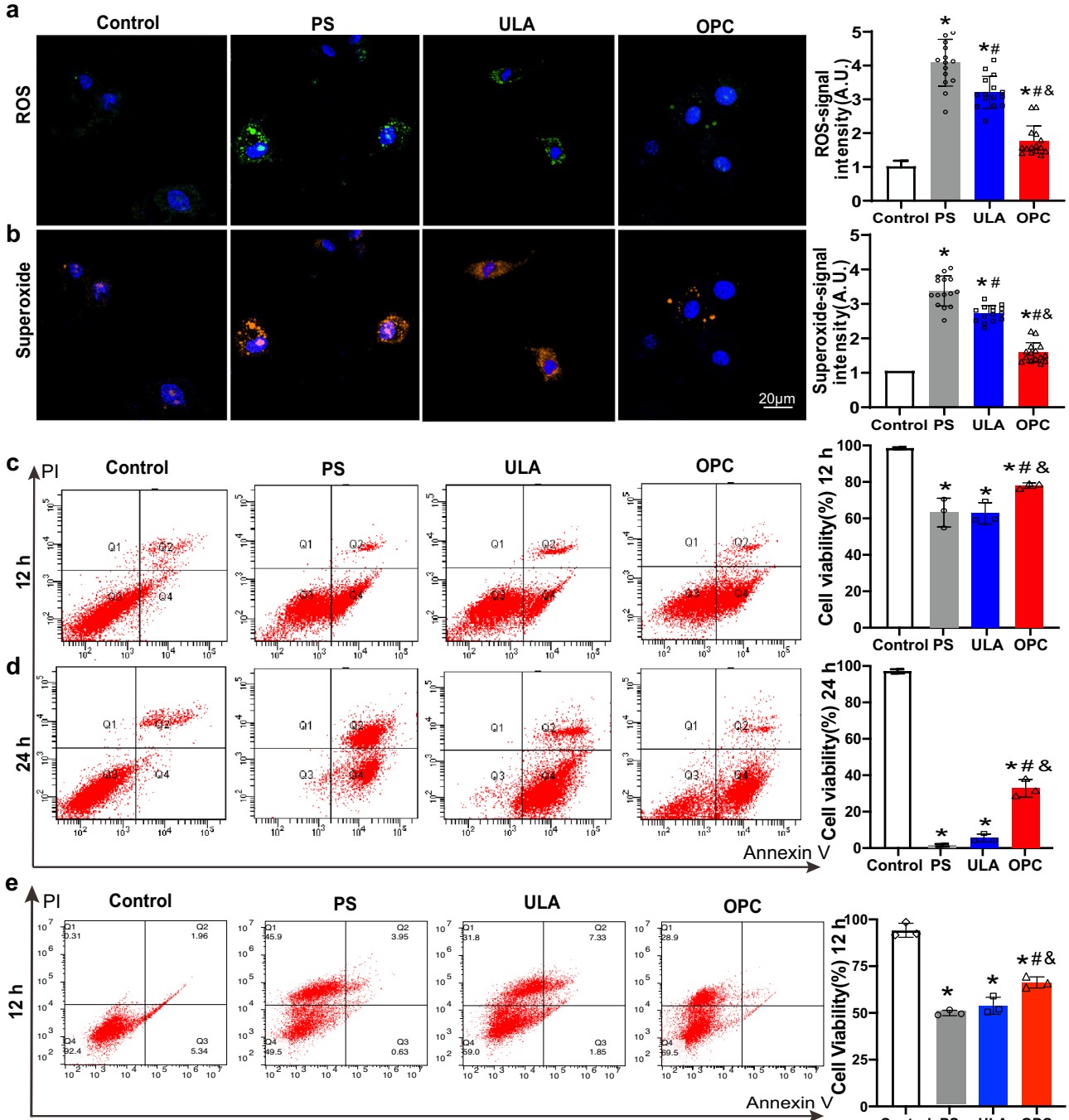

**Fig. 5 | Improvement in oxidative stress resistance in OPC-CSps following $H_2O_2$-induced injury.** Representative images and corresponding quantitative results of the (**a**) ROS and (**b**) superoxide fluorescence intensity following 24 h of $H_2O_2$ stimulation ($n = 15$ images from 3 experiments). The percentages of viable cells from each group following (**c**) 12 h and (**d**) 24 h $H_2O_2$ stimulation were determined by Annexin V/PI flow cytometry analysis ($n = 3$ biologically independent samples). **e** The percentages of viable cells from each group following 12 h of TNF-α stimulation were determined by Annexin V/PI flow cytometry analysis ($n = 3$ biologically independent samples). All data are shown as the mean ± SD, *$P < 0.05$ vs. Control, #$P < 0.05$ vs. PS, &$P < 0.05$ vs. ULA, one-way ANOVA.

transplanting OPC-CSps could achieve satisfactory long-term cardiac function recovery by effectively reducing myocardial inflammation and promoting angiogenesis.

Furthermore, protecting normal cardiac structure was a key issue for maintaining MI cardiac function. Owing to the severe inflammation and the hostile environment following MI, excessive degradation or impaired synthesis of ECM after MI was considered to accelerate ventricular remodeling, including myocardial fibrosis and cardiac hypertrophy, and ultimately lead to heart failure[54]. Cardiac fibrosis greatly reduces cardiac function by replacing necrotic myocardial tissue with

enlarged scars. In addition, massive cell death, a decrease in contractile activity in the affected zone, and increased hemodynamic burden were assumed to be the main causes of cardiac hypertrophy[55]. In this study, transplantation of OPC-CSps significantly decreased the infarct area, and an increase in viable cardiac tissue was observed (Fig. 7a, c, d), showing a beneficial effect in preventing cardiac hypertrophy (Fig. 7i−l). These results supported that transplanting OPC-CSps could greatly protect MI cardiac function by reducing ventricular remodeling.

In summary, to improve the MI therapeutic potential of CSps, a novel cell culture substrate, liquid crystal OPC, was prepared for

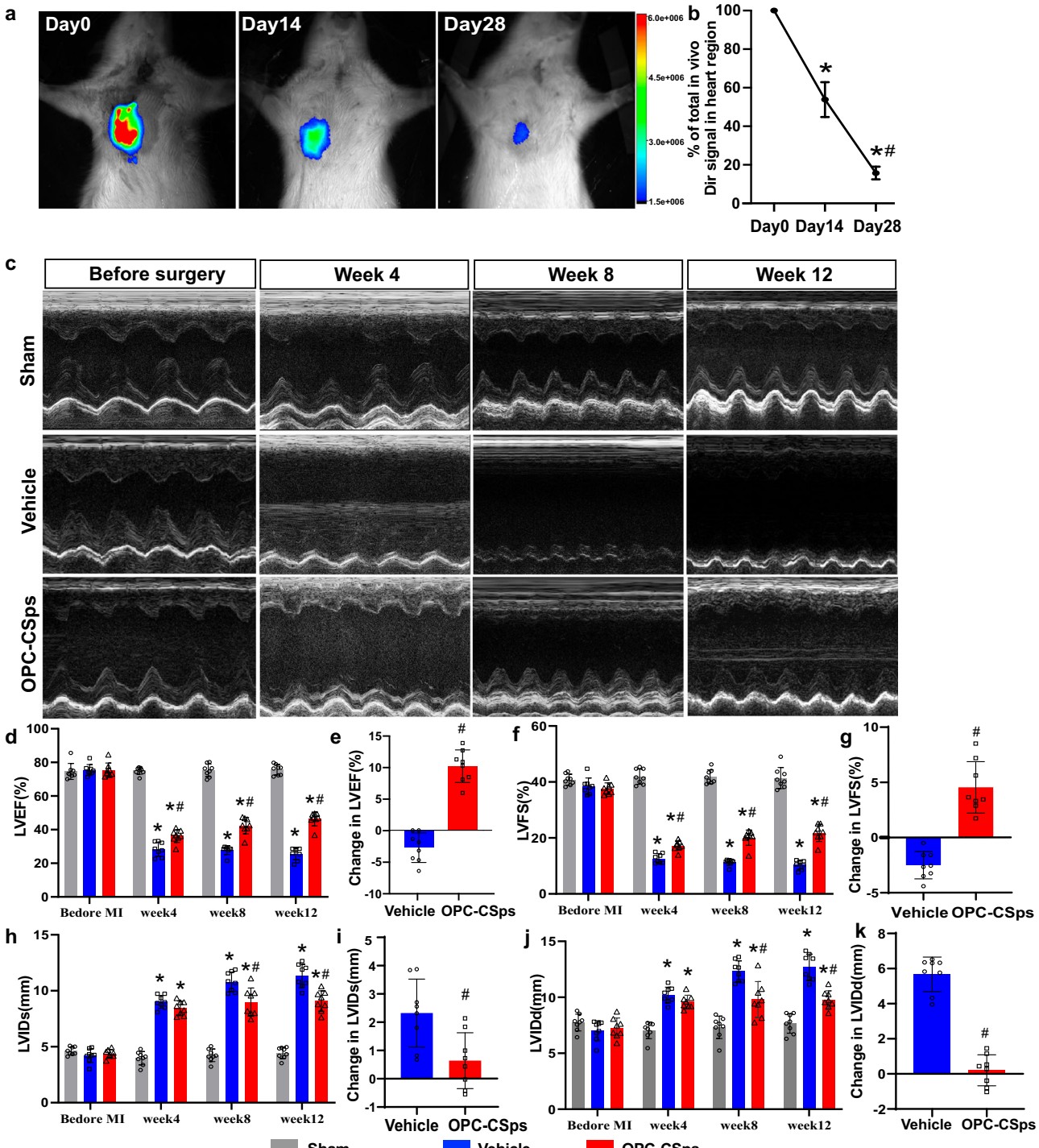

**Fig. 6 | OPC-CSps improved long-term MI cardiac functions. a** Representative image of DiR-labeled OPC-CSps at the instant, 14th, and 28th day after transplantation. **b** Quantification results of the survival of OPC-CSps in vivo at 28 days. (*$P < 0.05$ vs. D0, #$P < 0.05$ vs. D14, $n = 8$ rats). **c** Representative M-mode echocardiography images of the sham group, the vehicle group, and the OPC-CSps group. **d** LVEF of each group over 12 weeks. **e** The relative changes in LVEF at week 12 relative to week 4. **f** LVFS of each group over 12 weeks. **g** The relative changes in LVFS at week 12 relative to week 4. **h** The change in LVIDs over 12 weeks. **i** The relative changes in LVIDs at week 12 relative to week 4. **j** The change in LVIDd over 12 weeks. **k** The relative changes in LVIDd at week 12 relative to week 4. All data are presented as the mean ± SD, *$P < 0.05$ vs. sham, #$P < 0.05$ vs. vehicle, Two-sided student's t test (**e, g, i, k**), one-way ANOVA (**b**) or two-way ANOVA (**d, f, h, j**). For (**d–k**), $n = 8$ rats.

CSps culture and preconditioning. The OPC substrate served as a special mechanical cue to effectively promote the formation of CSps of controllable size. Furthermore, the OPC-CSps exhibited significant enhancement of biological function and antioxidative stress abilities. In the rat MI model, OPC-CSps not only showed great cell retention and survival in the infarct area but also significantly improved cardiac wall thickness, angiogenesis, and long-term cardiac function. In conclusion, using the OPC substrate could satisfy the demand for large-scale CSps production with excellent cardiac regeneration abilities for MI therapy.

 

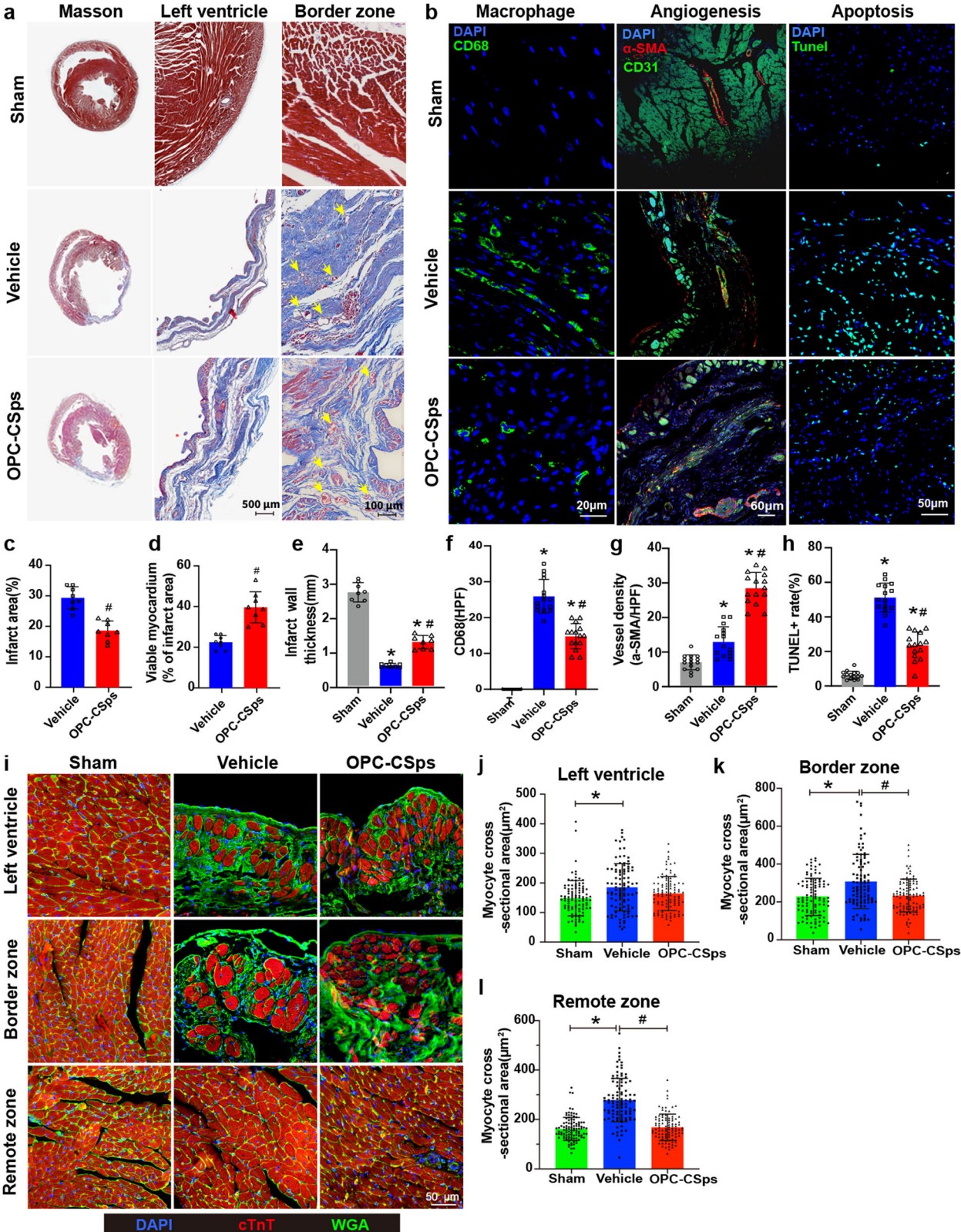

## Methods

### Animals

Rats were housed in specific pathogen-free conditions with 12 h day/light cycles. Rats were healthy and had free access to water and food. All animal studies were performed in accordance with the ethical guidelines of the National Guide for the Care and Use of Laboratory Animals and approved by Jinan University Animal Care and Use Committee (Approval numbers: IACUC-20210113-06). Four-week-old male *Sprague Dawley rats* (Guangdong Medical Laboratory Animal Center) were used for isolating primary CDCs, and 3-month-old female Sprague Dawley (Vital River) rats were used to establish myocardial infarction animal models. Every attempt was made to minimize the use of animals and pain.

**Fig. 7 | The reduction in cardiac inflammation, apoptosis, and hypertrophy after OPC-CSps transplantation. a** Representative images of Masson trichrome staining 12 weeks following MI, and yellow arrows mark the blood vessels in the border of the infarct area. ($n = 8$ rats). **b** Representative images of CD68$^+$ macrophages in the peri-infarct zone 2 weeks following MI, α-SMA$^+$ CD31$^+$ vessels in the infarct zone 12 weeks following MI, and TUNEL$^+$ cells in the peri-infarct zone 12 weeks following MI ($n = 15$ images from 3 rats). Spilt channels of the α-SMA and CD31 staining results are shown in Supplementary Fig. 4 ($n = 15$ images from 3 rats). **c** Quantitative results of the infarct area ($n = 8$ rats). **d** The percentage of viable myocardium at the infarct area ($n = 8$ rats). **e** Quantitative results of infarct wall thickness ($n = 8$ rats). **f** The quantitative results of CD68$^+$ macrophages per HPF

assessed by ImageJ software V1.8.0.112. HPF high-power field ($n = 15$ images from 3 rats). **g** The quantitative results of vessel density of each group ($n = 15$ images from 8 rats). **h** The quantitative results of the TUNEL$^+$ rate assessed by ImageJ software V1.8.0.112 ($n = 15$ images from 8 rats). **i** Representative images of WGA staining of heart tissues shown in different regions at 12 weeks. The cardiomyocyte membrane was stained with WGA (green), cardiomyocytes were identified by staining for cTnT (red), and DAPI showed nuclei. Quantitative analysis of cardiomyocyte cross-sectional area from **j** the left ventricle, **k** the border zone, and **l** the remote zone ($n = 15$ images from 8 rats). Each data point is represented as the mean ± SD, *$P < 0.05$ vs. sham, #$P < 0.05$ vs. vehicle, Two-sided student's $t$ test (**c, d**) or one-way ANOVA (**e, f, g, h, j, k, l**).

## Preparation of the octyl hydroxypropyl cellulose ester (OPC) substrates

OPC was prepared via esterification between *hydroxypropyl cellulose* (HPC) (Sigma–Aldrich, Mw = 100,000 g/mol) and *octanoyl chloride* (OC) (Sigma–Aldrich). Briefly, 5.0 g HPC was dissolved in 30 mL dehydrated acetone with mild stirring. Seven milliliters OC was added to the solution when HPC dissolved completely, and the reaction was kept at 55 °C for 4 h. Then, 300 ml distilled water was added to the reaction mixture, and a cream color sticky mass was obtained after removing the liquid phase. The cream-colored sticky mass was dissolved in acetone and precipitated by adding water to the solution. This step was repeated 6 times. After dissolving in ethanol and dialyzing in distilled water 15 times to remove the residual OC, OPC could be obtained after precipitation. Finally, the OPC product was dried in a vacuum at 55 °C for 48 h.

For the preparation of the OPC substrate, a 3% OPC concentration mixture was obtained by stirring OPC and ethanol for 1 h at 20 °C. It was cast onto clean culture dishes. After the solvents evaporated at room temperature, the dishes were washed with distilled water 10 times for 4 h each time. Then, the dishes were sterilized by Co60 irradiation (15kGy).

## Characterization of OPC

The surface characteristics of OPC were observed by polarized optical microscope (Carl Zeiss Axioskop40). An atomic force microscope (BENYUAN) was used to analyze the surface roughness in on-contact mode. The measurement of the water contact angle on the OPC substrate was tested at room temperature by a contact angle meter (Kruss DSA100) with ultrapure water as the testing liquid and a humidity of 80%.

The changes in the OPC structure when subjected to a shear force were tested by X-ray diffraction (XRD, Dmax1200). Briefly, 3% OPC solution was added to the glass surface and then covered with the magnesium sheet. Next, the samples were dried by placing them in a vacuum at 55 °C for 48 h. Then, the magnesium sheet was slid by weights of equal mass with a distance of 3 mm. The XRD patterns were recorded from 5° to 40° at a step width of 0.02° and scanning speed of 8°/min.

The crystallinity index (*Cr. I*) of OPC was determined according to the Segal method and calculated using Eq. (1)[56]

$$Cr.I = \frac{I_{002} - I_{amorph}}{I_{002}} \tag{1}$$

where $I_{002}$ is the maximum intensity of the main diffraction, and $I_{amorph}$ is the intensity of the amorphous background scatter measured at $2\theta = 18°$ where the intensity is minimum.

The crystallite diameter ($D_{002}$) perpendicular to the (002) plane was calculated from the Scherrer Eq. (2)[57]

$$D_{002} = \frac{k\lambda}{\beta \cos \theta} \tag{2}$$

where $K$ is a Scherrer constant that equals 0.9, $\lambda$ is the wavelength of the radiation (1.54 Å for CuKα), $\beta$ is the width of the peak at half maximum, and $\theta$ is the angle of incidence.

Finally, the rheological properties of OPC were measured by a DHR-2 stress-controlled rheometer (TA Instruments). The oscillation-frequency mode at 37 °C was incorporated for rheological tests with a strain of 1% and $\omega = 1–100$ rad s$^{-1}$.

For the OPC toxicology test, $2 \times 10^3$ CDCs were seeded in 96-well plates and cultured in 100 μl of CSps culture medium or OPC immersion culture medium. Ten microliters of CCK8 (Dojindo) was added to each well and incubated for 2 h every 24 h for five consecutive days, and the optical density values were recorded at 450 nm wavelengths.

## Isolation and culture of CDCs

Cardiac tissue specimens from the septum of the left ventricle were minced and digested with collagenase IV (Sigma) for 20 min at 37 °C. These tissues were plated on poly-d-lysine (Sigma)-coated dishes in CSps culture medium, which consisted of 500 ml Iscove DMEM (Corning), 1% L-glutamine (Corning), 1% penicillin–streptomycin solution (Corning), 10% fetal bovine serum (BD Bioscience) and 0.1 mmol/L β-mercaptoethanol (Gibco). After 1–2 weeks, a monolayer of adherent cells that grew out from these tissues was harvested by 0.05% trypsin and passaged on poly-d-lysine-coated dishes. Following 3–7 days of cultivation, the CSps were collected and plated onto fibronectin-coated dishes and expanded as monolayer CDCs. All cultures were cultured in 5% CO$_2$ at 37 °C.

## Preparation of CSps

Polystyrene (PS) substrate, ultralow attachment (ULA) substrate, and OPC substrate were used to culture CDCs and obtain CSps. Cells were seeded onto three different substrates at a density of $7 \times 10^4$ cells/cm$^2$. The formation of CSps in each group was observed by an inverted microscope (Olympus IX71) for 5 consecutive days. The diameter and the total number of CSps at the same time point were measured using ImageJ software V1.8.0.112, and at least 30 CSps from each group were randomly chosen.

## Flow cytometry test

The expression levels of the CSps surface markers CD31 (1:200, GB11063-3, Servicebio), CD34 (1:200, ab81289, Abcam), CD90 (1:200, ab225, Abcam), CD105 (1:200, ab156756, Abcam), Sca-1 (1:200, ab51317, Abcam), and KDR (1:200, sc6251, Santa Cruz) were determined by flow cytometry. After a 3-day cultivation, cells and CSps from different substrates were obtained and digested into single cells. After incubation with the primary antibodies for 1 h and the corresponding secondary antibodies for 30 min, Alexa Fluor 488 goat anti-mouse IgG (1:200, ab150113, Abcam) or Alexa Fluor 488 goat anti-rabbit IgG (1:200, ab150077, Abcam) was used. The staining results were analyzed by flow cytometry (BD FACSCanto), and a negative isotypic control was used during the analysis.

## Ultrastructure analysis

The cells and CSps from each substrate were harvested and fixed with 2.5% glutaraldehyde overnight at 4 °C. Following dehydration with a series of graded ethanol solutions, samples were immersed in 1% osmium tetroxide for 2 h. Next, the samples were embedded in resin and cut to 60 nm thickness. Then, the cell ultrastructure was visualized and photographed with transmission electron microscope (TEM, JEOL).

## Cell proliferation, apoptosis, and enzyme-linked immunosorbent assay (ELISA)

Cell proliferation was determined by CCK8 (Dojindo) after being cultivated for 3 days on each substrate. Cell apoptosis rates were determined by using the Alexa Fluor® 488 Annexin-V/Dead Cell Apoptosis Kit (Elabscience) according to the manufacturer's instructions. The results were analyzed and recorded by flow cytometry (BD FACS-Canto). Survival cells in the Q4 gate were quantified and analyzed. Following 3 days of cultivation, the culture supernatants from each group were harvested, and the $IL$-$1\beta$ concentration was measured by rat $IL$-$1\beta$ ELISA kits (MEIMIAN).

## Real-time quantitative PCR (qRT-PCR)

Total RNA was extracted using TRIzol reagent. High-Capacity cDNA Reverse Transcription Kits (Thermo Fisher Scientific) were used for cDNA synthesis according to the manufacturer's instructions. qRT-PCRs were performed on a Mini Cycler PCR instrument with SYBR Green reagent (Toyobo). Gene expression was normalized to $GAPDH$ mRNA, and relative expression was calculated by $2^{-\Delta\Delta CT}$. The primer sequences of the detected genes are listed in Supplementary Table 1, and the primers were synthesized by Guangzhou Generay Biotechnology Company.

## Western blot analysis

The cells and CSps cultured on the PS, ULA, and OPC substrates were lysed in RIPA buffer (Solarbio) for protein extraction. The protein concentrations of the samples were adjusted by bicinchoninic acid (BCA) protein assay (Thermo Fisher Scientific) according to the manufacturer's protocol. Forty micrograms of protein was loaded into each lane of a 10% sodium dodecyl sulfate–polyacrylamide gel electrophoresis (Millipore) gel and transferred to polyvinylidene fluoride membranes (Millipore). After being blocked with 5% bovine serum albumin in Tris buffered saline Tween (Biosharp) at room temperature for 1 h, the membranes were incubated with the primary monoclonal antibodies against $caspase$-$1$ (1:500, ab1872, Abcam) and $\beta$-$actin$ (1:1000, ab8226, Abcam) in TBST overnight at 4 °C, after which HRP-labeled Anti-Rabbit IgG antibody (1:2000, 7074, Cell Signaling Technology) and HRP-labeled Anti-mouse IgG antibody (1:2000, 7076, Cell Signaling Technology) was added and incubated for another 1 h. The results were visualized via enzyme-linked chemiluminescence by an ELC kit (Thermo Fisher Scientific). $\beta$-$Actin$ was used as an internal control.

## Metabolic analysis of CSps

The glucose uptake ability of CDCs was evaluated by using the fluorescent glucose 2-NBDG (Cayman). After 3 days of cultivation of CDCs on different substrates, all the culture medium was removed and replaced with glucose-free DMEM containing 50 μM 2-NBDG for 30 min. Consequently, the fluorescence intensity of the cells was measured by flow cytometry (BD FACSCanto). Following a 3-day cultivation, the culture medium of each group was collected to test the extracellular lactate contents, and the reagent kit was the Lactate Colorimetric Assay Kit (Nanjing JianCheng).

For the intracellular ATP content assay, CDCs were cultured on different substrates for 3 days and formed CSps, and single cells were isolated from the CSps of each group and seeded in a 96-well plate at a density of $1 \times 10^5$. The intracellular ATP content was determined by using the CellTiter-Glo Luminescent Cell Viability Assay (Promega). For the mitochondrial membrane potential measurement, digested cells were washed twice with PBS and incubated with 2 μM Rho123 (Beyotime) for 20 min in a dark environment at 37 °C. Fluorescence images were taken under a fluorescence microscope (Olympus, FV3000: Olympus FV31S-SW software displayed), and the fluorescence intensity of the cells was analyzed with ImageJ software V1.8.0.112.

## RNA-Seq analysis

According to the manufacturer's protocol, a total RNA Kit I (Omega) was used to extract sample RNA. Sequencing was performed on the Illumina platform. The raw RNA-seq reads were aligned to the Rattus norvegicus genome (rn6) by hisat2 (version:2.1.0). Mapped reads were counted by featureCounts (v.1.6.2), and gene expression was calculated by R and the DESeq2 package. Significant differentially expressed genes (DEGs) among cells cultured on the PS, ULA, and OPC substrates were evaluated using DESeq (1.28.0), and genes with a |log2FC| ≥ 1 and an adjusted $P$ value ≤ 0.05 were selected for further analysis. Hierarchical clustering was performed for DEGs using a heatmap. Kobas (3.0) was used for Kyoto Encyclopedia of Genes and Genomes (KEGG) pathway enrichment analysis. Additionally, gene set enrichment analysis (GSEA) was carried out using GSEA v.4.2.3.

## Oxidative stress resistance level measurement

Following a 3-day cultivation, CDCs formed CSps on different culture substrate, and the CSps were subjected to $H_2O_2$ stimulation for 24 h. And then singles cells were isolated from the CSps and used for intracellular ROS and superoxide production measurement with the cellular ROS/superoxide detection assay kit (Abcam). In short, the isolated cells were incubated for ROS/superoxide detection for 30 min at 37 °C. The results were observed and recorded by a laser confocal scanning microscope (Olympus, FV3000: Olympus FV31S-SW software displayed). In addition, after the CSps were exposed following 12 h or 24 h of exposure to $H_2O_2$, CSps were digested into single cells and used to measure the cell survival rate with the Alexa Fluor® 488 Annexin-V/Dead Cell Apoptosis Kit (Invitrogen). The results were analyzed by flow cytometry (BD FACSCanto).

## Anti-inflammation ability measurement

Following a 3-day cultivation, CDCs formed CSps on different culture substrate, and the CSps were subjected to 50 ng/ml TNF-α stimulation for 12 h. And then singles cells were isolated from the CSps and used to measure the cell survival rate with the Alexa Fluor® 488 Annexin-V/Dead Cell Apoptosis Kit (Invitrogen). The results were analyzed by flow cytometry (BD FACSCanto).

## Rat MI model and CSps transplantation

The establishment of the MI model and transplantation methods were carried out. In brief, female rats aged 3 months (220 ± 20 g) were anesthetized with 3% isoflurane and ventilated through endotracheal intubation. Mechanical ventilation was provided with room air at 60 to 70 breaths min$^{-1}$ using a rodent respirator (Taimeng Company). Subcutaneous stainless-steel electrodes were used to record the standard electrocardiogram. After shaving the chest, a left thoracotomy was performed to expose the heart at the fifth intercostal space. The left anterior descending coronary artery (LAD) was ligated using a 6-0 silk suture, and ischemia was confirmed by observing ST segment (or J point) elevation and the occurrence of cardiac cyanosis. After stabilizing for 15 min, rats in the vehicle group were transplanted with blank Matrigel, and rats in the OPC-CSps group were transplanted with OPC-CSps Matrigel suspensions (0.1 mL). OPC-CSps were transplanted into the OPC-CSps group with a total number of $5 \times 10^6$ cells. For the sham group, rats were subjected to the same procedure without ligation and Matrigel injection. Finally, the animals were closed at the chest and

monitored, and antibiotics and 0.9% normal saline solution were administered. Three rats from each group were killed at week 2 and cardiac samples were collected to assess macrophage infiltration. Continuous cardiac cycles were collected for 8 rats from each group.

## Ex vivo imaging analysis

Prior to transplantation, CSps were labeled with 3.5 µg/ml 1,1-dioctadecyl-3,3,3,3-tetramethylindotricarbocyanine iodide (DiR, Invitrogen) following the manufacturer's instructions. The survival status and localization of Dir-labeled CSps were observed using a Bruker In Vivo Xtreme II Imager (Bruker). The Bruker Molecular Imaging Software (IB5438150 Rev. B 12/12) and ImageJ software V1.8.0.112 were applied for imaging processing and data analysis.

## Echocardiogram

Cardiac function and the movement of the left ventricular wall were measured using a Vevo 2100 ultrasonic system before surgery and at 4, 8, and 12 weeks postsurgery. After the rats were anesthetized with isoflurane, the four-chamber and two-chamber sections of the left ventricle were obtained by ultrasound, left ventricular end-diastolic volume (LVEDV) and end-systolic volume (LVESV) were measured using the trackball at the 4-chamber and 2-chamber sections of the left ventricle, and the left ventricle ejection factor (LVEF) was calculated by the equation: LEVF= (LVEDV-LVESV)/LVEDV*100%.

## Histology and immunochemistry assay

For cell sample preparation, CSps from each group were harvested, fixed in a 75% ethanol solution, and embedded in OCT compound, and 5µm sections were used. For tissue sample preparation, after rats were euthanized, the heart tissues were harvested, fixed in 4% paraformaldehyde, embedded in paraffin, and serially sectioned at 5µm thickness. Masson trichrome staining was performed on tissue sections. For immunofluorescence staining, the sections were blocked with 5% goat serum and then incubated with primary antibodies, including anti-caspase-1(1:100, ab1872, Abcam), anti-CD68 (1:200, 360018, Zhengneng), anti-cardiac troponin T (cTnT, 1:500, ab209813, Abcam), anti-smooth muscle alpha-actin (α-SMA, 1:300, 41550, SAB), and anti-CD31 (1:300, GB11063-3, Servicebio) at 4 °C overnight. After washing with PBS 3 times, 488 nm goat anti-rabbit (1:400, ab150077, Abcam), 594 nm goat anti-rabbit (1:400, ab150080, Abcam), 488 nm goat anti-Mouse(1:400, ab150113, Abcam), or 594 nm goat anti-Mouse (1:400, ab150116, Abcam) secondary antibodies were added and incubated for 2 h at room temperature. The dilutions were 1:200 for the primary antibody and 1:400 for the secondary antibody. Apoptosis and the cross-sectional area of myocardial cells were assessed by terminal deoxynucleotidyl transferase-mediated dUTP nick end labeling (TUNEL) (Promega) and FITC-labeled wheat germ lectin (WGA) (1:500, Thermo Fisher Scientific) staining, respectively. DAPI showed the nuclei. All stained images were observed and photographed by laser confocal scanning microscope (Olympus, FV3000: Olympus FV31S-SW software displayed).

## Statistical analysis

All analyses were performed using GraphPad Prism 9.0 (GraphPad Software Inc). For comparisons of two groups, a two-sided Student's $t$ test was used. Comparisons of multiple groups were made using one- or two-way ANOVA. The KEGG pathway of genes were analyzed by one-side hypergeometric analysis with Benjamini−Hochberg multiple testing correction. All quantitative results were expressed as the mean ± standard deviation, and differences with a $P$ value < 0.05 were considered statistically significant.

## Reporting summary

Further information on research design is available in the Nature Portfolio Reporting Summary linked to this article.

## Data availability

The RNA-seq data from CDCs cultured on the PS, ULA, and OPC substrates used in this study are available in the GEO database under accession code "GSE223508". All other data supporting the findings of this study are available within the article and its supplementary files. Any additional requests for information can be directed to, and will be fulfilled by, the corresponding author(s). Source data are provided with this paper.

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

## Acknowledgements

This work was supported by the National Natural Science Foundation of China [grant number: 31771064,31800819]; National High Technology Research and Development Program for Young Scientists [Grant number 2014AA020534]; Natural Science Foundation of Guangdong Province [grant number: 2022A1515011085] and Science and Technology Planning Project of Guangzhou [grant number: 201904010137].

## Author contributions

Y.W.W.: Conceptualization, formal analysis, writing - original draft, review & editing. Q.L.: Conceptualization, formal analysis, writing - original draft. J.P.Z., D.X.W., and Y.L.Z.: Methodology. J.M.C., J.X.W., J.L., and J.L.L: Data curation. J.-H.Z.: Funding acquisition, Writing - review & editing. Z.W.: Conceptualization, writing - review & editing, supervision, funding acquisition.

## Competing interests

The authors declare no competing interests.
