## [Peer Review File · Nature Communications]

REVIEWER COMMENTS

Reviewer #1 (Remarks to the Author):

In this manuscript the authors propose to enhance the therapeutic potential of cardiospheres (CSps) and cardiosphere-derived cells (CDCs) following myocardial infarction. They propose that culture of the cardiospheres on a novel liquid crystal surface (OPC) would enhance the therapeutic potential. They culture cardiospheres on polystyrene, an ultralow attachment substrate (ULA) which is not defined and on the OPC surface. They assess cell viability and expression of relevant markers and test the therapeutic potential of the cells cultured on the OPC surface.

I am not a materials scientist and so cannot comment on the work to generate the liquid crystal surface.

A major shortfall in the manuscript is that, having compared the three cell types in a range of experiments in vitro, the in vivo assessment is a comparison of sham, vehicle and OPC-CSps so that no increased benefit of the novel culture is tested in vivo. Having checked a paper cited by colleagues of the authors (ref 24) I would suggest that the improvement in ejection fraction given in this manuscript was similar to that seen in that reference.

I have a number of other concerns about the work presented in this manuscript.

The authors discuss cardiosphere formation on the three surfaces. The details here are not very clear, the authors say that “The adherent CDCs on the PS substrate converged and formed regular spherical cloning, and fibroblast-like supporting cells were in the lowest of the PS-CSps”. It is not clear what the mean by ‘cloning’ or of the ‘lowest’ of the PS-CSps and they show no staining for fibroblast-like cells. They also say that the OPC-CSps show a ‘stable tendency’, does that refer to the fact that these spheres do not substantially increase in size?

The authors demonstrate increased expression of caspase-1 and IL1-beta in the OPC-CSps. It is not clear to me why this is an advantage. The authors link this activation of the inflammatory response to upregulation of stem cell markers and of growth factors but the two may result from independent effects of the OPC culture.

In figure 3, the authors measure proliferation but do not make it clear whether this is in CSps or in CDCs isolated from the CSps and seeded on the different surfaces. Subsequent experiments to measure mitochondrial function (Fig 4) and oxidative stress (Fig 5) refer to CDCs in the methods but CSps in the figure legend.

The results section of the RNA-seq data is confusing. The authors state that “the DEGs between the OPC-CSps and the ULA-CSps were enriched in the glycolytic/glycogenic pathway and the HIF-1 signaling pathway” implying that these pathways were higher in the OPC-CSps than the ULA-CSps. But then they say “Moreover, gene set enrichment analysis (GSEA) also revealed the downregulation of the hypoxic and glycolytic components in the OPC-CSps compared to the ULA-CSps”. They go on to show that glycolysis is lower in the OPC-CSps (or CDCs) than the other two groups which is interesting as normally stemness is associated with glycolysis and the authors have shown increased proliferation (associated with stemness) and stem cell genes in this population.

In the in vivo experiment, the authors again don't make it clear whether they inject CSps or CDCs. They refer to OPC-CSps Matrigel suspensions but then give a cell number which would be hard to determine from a sphere culture. In the ex vivo histology, they show a measurement of the ‘infarct wall’ in the sham, which has no infarct and the staining in Fig 7b suggests there are no major blood vessels in the sham heart, which is unlikely.

In the discussion on page 25, the authors refer to microenvironmental cues from ECM increasing expression of CD31 and CD34 but there is no discussion of ECM on the various surfaces. The discussion of the mitochondrial changes shows confused interpretation of these data. The authors say “In this study, the oxidative phosphorylation of OPC-CSps was enhanced, while the ULA-CSps altered their energy production toward glycolysis” and then they say “In addition, the OPC group showed a significant decrease in glucose uptake levels (Fig. 4f), suggesting that the OPC-CSps may survive longer in nutrient-limited conditions”. If oxidative phosphorylation is enhanced then the cells still require glucose, since low levels of other substrates are provided in the cell culture medium used. Furthermore, if cells are injected into the infarct zone, where oxygen supply is limited, then they will need to rely on glycolysis

Reviewer #2 (Remarks to the Author):

Dr. Li and colleagues have submitted a manuscript entitled « Mechanically induced pyroptosis enhanced cardiosphere oxidative stress resistance and metabolism for myocardial infarction therapy ». Briefly, the authors propose liquid crystal substrate, octyl hydroxypropyl cellulose ester (OPC) as an effective culture system for cardiospheres (CSps) derived from cardiosphere-derived cells. OPC substrate served as a special mechanical cue to effectively promote the formation of CSps of controllable size.

Moreover, the CSps obtained on OPC substrate showed improved cell bioactivities and oxidative stress resistance under stimulation of mechanical-induced pyroptosis. In a rat myocardial infarction model, OPC-CSps improved long-term cardiac function, promoted angiogenesis, and reduced cardiac remodeling. This is an interesting study. However, several points need to be clarified prior to publication. In the present version of the manuscript, the results were not clear enough to support author's conclusions.

-The authors mention (see line 127) fibroblast-like supporting cells with CSps but these cells are not visible in the Figure 2. Please include images with high magnification to appreciate these cells. What is the percentage of the fibroblast-like supporting cells? I suppose that these cells are included in the phenotypic analyses. In my opinion, the percentage of these cells would allow better interpretation of the results and should therefore be included in the manuscript.

- Please include images with high magnification to appreciate the morphological differences of CSps cultured on PS, ULA and OPC.

- How do the authors explain that the size of OPC-CSps remains stable during culture even though the cells proliferate more and the survival rate is high with OPC.

- The authors mention that the expression of caspase-1 was observed in the peripheral cells of the OPC-CSps. Thus, proliferative cells are found in the center of CSps? Immunostaining images showing cell proliferation should be included in the manuscript to better appreciate these results.

- Does H₂O₂ treatment increase the level of caspase-1 and IL-1 β level at the peripheral cells?

- Figure 5 needs to be revised. The images should illustrate CSps not separated cells. Idem for Figure 4j.

- In the manuscript, cardiac function was measured in post-MI rats by trans-thoracic echocardiography using two-dimensional parasternal long-axis method but the Figure 6 present M-mode results. It should be noted that in ischemic heart failure created by coronary artery ligation, it is not possible to properly evaluate cardiac function using M-mode.

- Measurement of cardiomyocyte size in transversal section is difficult. The most appropriate method is measurement in longitudinal planes combined with immunostaining of intercalated discs.

- Is it possible to visualize DiR-labelled cells to explore the phenotype of these grafted cells?

Reviewer #3 (Remarks to the Author):

In the present study, the authors investigated that OPC-CSps exhibited better anti-oxidative stress, anti-inflammatory and pro-angiogenic effects in acute MI. The manuscript is well structured, but still requires substantive modifications, otherwise it will be rejected

1. In the article, the author stated that the peripheral cells of OPC-CSps in the PS group, ULA group and OPC group showed pyroptosis positive. Has the author tested and compared the expression of other types of death methods such as necroptosis, apoptosis or ferroptosis? Why do research on pyroptosis directly.

2. The KEGG results in Figure 4c show that the HIF-1 signaling pathway is enriched, whether to verify the activation of this pathway in vivo and in vitro.

3. The author stated that oxidative stress injury and inflammation play a key role in acute MI, but in vivo experiments only verified the ability of OPC to resist oxidative stress injury, and its anti-inflammatory effect in vitro did not appear to be superficial.

4. In Figure 7b, the author only uses the amount of CD68 to represent inflammation, whether it can represent the polarization of macrophages, and to improve the effect of OPC-CSps on the polarization of macrophages in vitro.

5. In Figure 7b, the author only shows blood vessels by single staining of α -SMA. Is it not rigorous enough? It is recommended to co-stain with endothelial cell indicator CD31, and double positives indicate blood vessels.

6. The author stated in the previous article that OPC has the activation of pyroptosis, whether the expression level of pyroptosis is expressed in the subsequent in vivo and in vitro verification, and whether OPC-CSps has an impact on pyroptosis.

7. The author should add PS and ULA groups for comparison in the verification of in vivo, not just in vitro.

RESPONSES TO REVIEWERS' COMMENTS

We appreciate your prompt reviews and valuable suggestions, and we had revised the manuscript according to your comments and suggestions. All revised text was highlighted in yellow color in the revised manuscript, and the grammar and spell error were showed in red color.

Response to reviewer 1:

1. A major shortfall in the manuscript is that, having compared the three cell types in a range of experiments in vitro, the in vivo assessment is a comparison of sham, vehicle and OPC-CSps so that no increased benefit of the novel culture is tested in vivo. Having checked a paper cited by colleagues of the authors (ref 24) I would suggest that the improvement in ejection fraction given in this manuscript was similar to that seen in that reference.

Author Answer: Thank you for your comments. Our previous study (ref24) evidenced that using pericardial fluid to simulate the in vivo inflammatory environment for CSps could improve cell viabilities, and it explore a novel cell delivery route as pericardial cavity implantation for MI therapy. On this base, the aim of the present study was to explore a stable, convenient and efficient precondition treatment for preparing CSps. The results showed that using OPC substrate could provide unique mechanical induction and efficiently acquire adequate size-control CSps with improved cell bioactivities.

According to the reviewer's comment, it is reasonable to compare the CSps prepared by different approaches and observed their therapeutic effects on MI, but in accordance to the 3R principle of animal experiments (reduction, replacement and refinement), the present study verified the functions of CSps (CDCs) among different groups in vitro before performing the in vivo studies. The in vitro studies evidenced that CDCs from the OPC-CSps showed improvements of cell survival, proliferation ability, and growth factors paracrine levels. Also, improved mitochondrial function and enhancement of oxidative stress resistance of the CDCs from OPC-CSps were observed. In addition, we replenished the anti-inflammatory experiments in the revised manuscript, and the result showed that CDCs from OPC-CSps had the best cell viabilities when compared to the cells from PS-CSps and ULA-CSps. Taken the in vitro experiments results together, OPC-CSps showed superior bioactivities compared to the PS-CSps and ULA-CSps, so the OPC-CSps was further examined its therapeutic effects on MI rats.

Although the left anterior descending coronary artery (LAD) was used to establish the rat MI model in both studies, the significance of the cardiac function indicators improvements would be more reasonable when compared in the same batch of the animal. Moreover, compared to the previous in vivo study, we did a longer

observation following the transplantation of OPC-CSps (12 weeks), and the results showed increase of the cardiac function improvements were observed in the OPC-CSps group, which showed important long-term therapeutic effects of OPC-CSps.

The corresponding revisions are as follows:

Results

OPC-CSps showed enhanced oxidative stress resistance and anti-inflammatory effect

H₂O₂ stimulation was used to test cellular oxidative stress resistance. As shown in Figure 5a and 5b, after exposure to H₂O₂ for 24 h, the fluorescence intensity of the ULA group was significantly lower than that of the PS group ($P < 0.05$). Moreover, the fluorescence intensity of the OPC group further decreased compared to the PS group and the ULA group, suggesting that OPC-CSps generated the least reactive oxygen species (ROS) and superoxide under an oxidative stress environment. Following 12 h of H₂O₂ stimulation, the percentage of viable cells in the OPC group ($73.7\% \pm 1.4\%$) was significantly higher than that in the PS group ($59.7\% \pm 7.5\%$) and the ULA group ($59.2\% \pm 5.4\%$) ($P < 0.05$) (Fig. 5c). There was no difference in the cell survival rate between the PS-CSps and the ULA-CSps. Following 24 h of H₂O₂ stimulation, a similar trend in the survival rate was observed among the three groups, and the survival rates of the PS, ULA, and OPC groups were $1.5\% \pm 0.7\%$, $5.4\% \pm 2.1\%$, and $31.7\% \pm 4.7\%$, respectively (Fig. 5d). The anti-inflammatory effects were also examined. Following 12 h of TNF- α stimulation, the survival rates of the PS, ULA, and OPC groups were $50.0\% \pm 3.7\%$, $53.8\% \pm 4.5\%$, and $66.3\% \pm 3.0\%$, which were significantly lower than the control group, while the cells from the OPC-CSps group exhibited the best anti-inflammatory effect among 3 groups (Fig. 5e).

Figure 5. Improvement in oxidative stress resistance in OPC-CSps following H₂O₂-induced injury. **a-b** Representative images and corresponding quantitative results of the (a) ROS and (b) superoxide fluorescence intensity following 24 h of H₂O₂ stimulation (*n* = 15 images from 3 experiments). **c-d** The percentages of viable cells from each group following (c) 12 h and (d) 24 h H₂O₂ stimulation were determined by Annexin V/PI flow cytometry analysis (*n* = 3). **(e)** The percentages of viable cells from each group following 12 h of TNF-α stimulation were determined by Annexin V/PI flow cytometry analysis (*n* = 3). All data are shown as the mean ± SD, **P* < 0.05 vs. Control, #*P* < 0.05 vs. PS, & *P* < 0.05 vs. ULA, one-way ANOVA.

Methods and materials

Anti-inflammation ability measurement

Following a 3-day cultivation, CDCs formed CSps on different culture substrate, and the CSps were subjected to TNF-α stimulation for 12 h. And then singles cells were isolated from the CSps and used to measure the cell survival rate with the Alexa Fluor® 488 Annexin-V/Dead Cell Apoptosis Kit (Invitrogen). The results were analyzed by flow cytometry (BD FACSCanto).

2. The authors discuss cardiosphere formation on the three surfaces. The details here are not very clear, the authors say that “The adherent CDCs on the PS substrate converged and formed regular spherical cloning, and fibroblast-like supporting cells were in the lowest of the PS-CSps”. It is not clear what the mean by ‘cloning’ or of the ‘lowest’ of the PS-CSps and they show no staining for fibroblast-like cells. They also say that the OPC-CSps show a ‘stable tendency’, does that refer to the fact that these spheres do not substantially increase in size?

Author Answer: Thank you for your comments, they are very important for improving the quality of our manuscript.

Firstly, we are very sorry for the ambiguous expressions in the manuscript. “The regular spherical cloning” in the manuscript refers to the spherical aggregates formed on PS substrate, and the fibroblast-like cells refers to the mesenchymal-like cells around the CSps. Cardiospheres (CSps) are isolated and cultured from the heart tissues and they contained a mix population of cells expressed related markers, including cardiac progenitor cells (CPCs), endothelial cells, and mesenchymal cells. You are right about that we should identify the sub-cell types of the CSps, so we performed the flow cytometry analysis to verify the portions of different cell population of the CSps in the present study, and CD90 and CD105 was typical mesenchymal cell markers of the CDCs [ref1]. According to your comment, we have revised the writing in the manuscript for better illustration, and we have also replenished high magnification fields in the supplementary materials to better show the formation process of the CSps on different substrates, where mesenchymal-like cells and other supporting cells could be clearly seen around the PS-CSps.

Secondly, you are right about that the description of “OPC-CSps show a ‘stable tendency’” refers to these spheres do not substantially increase in size. And we realized that the original expression may lead to misunderstanding, and we have also revised the corresponding expressions in the manuscript. **For your convenience, the corresponding revisions are listed as follows:**

Results

The OPC substrate promoted CSps formation and progenitor phenotypes

The formation of CSps on the polystyrene (PS) substrate, the ultralow attachment (ULA) substrate and the OPC substrate was observed, and different shapes of CSps were acquired (Fig. 2a and Supplementary Fig.1). The adherent CDCs on the PS substrate converged and formed regular spherical aggregates, and supporting cells surrounded at the base of the PS-CSps. The ULA-CSps were formed by suspended single-cell stacking, and they developed into noncircular, oval, and irregular shapes. On the OPC substrate, CDCs initially attached to the substrate and gradually aggregated to form regular and circular spheroids, and no supporting cells were observed around the OPC-CSps. During the formation of CSps, the sizes of OPC-CSps showed a stable growth tendency compared with the obvious increase of the PS-CSps and the ULA-CSps (Fig. 2b and Supplementary Fig.1). Following a 3-day cultivation, the sizes of PS-CSps, ULA-CSps, and OPC-CSps were $280.96 \pm 40.56 \mu\text{m}$,

203.34 ± 69.36 μm, and 120.29 ± 15.34 μm, respectively. In addition, the spheroid density on the OPC substrate was significantly higher than that on the other two substrates (Fig. 2c). The expression levels of CSps surface markers were analyzed. Compared to the PS group and the ULA group, the OPC group exhibited the highest expression of KDR and Sca-1 ($P < 0.05$). In addition, the ULA group and the OPC group showed an increase in the expression of CD31 and CD34 ($P < 0.05$), along with a decrease in CD90 ($P < 0.05$) and no significant difference in CD105 compared to the PS group (Fig. 2d).

Supplementary Fig.1. Morphological changes in CSps on the PS, ULA, and OPC substrates in high magnification fields.

[ref1] Darryl R. Davis, Yiqiang Zhang, Rachel R. Smith, et al., Validation of the Cardiosphere Method to Culture Cardiac Progenitor Cells from Myocardial Tissue. PloS One. 2009, 4(9): e7195. (citation frequency: 240)

For your convenience, the related materials in the ref1 are listed as follows:

Adult human cardiospheres exhibit a complex mixed cell phenotype

As shown above, the direct outgrowth from cardiac samples contains a mixed cell population of CPCs (c-Kit), endothelial cells (CD31, CD34) and mesenchymal cells (CD90). To explore the effects of three dimensional sphere culture on these sub-populations, we examined human cardiospheres for expression of CPC-related proteins (c-Kit, MDR1), mesenchymal-related proteins (CD105, CD90, procollagen type I, vimentin), endothelial-related proteins (CD105,

CD133, CD31, CD34), hematopoietic-related proteins (CD34, CD45), cardiomyocyte-related proteins (Cx43, cMHC, cTnI, Nkx2.5 and desmin), and Ki67 to identify proliferative cells. As shown in Figure 5, cardiospheres consist of distinct layers with expression of the CPC marker c-Kit in the core (Figure 5A) and the mesenchymal marker CD105 on their periphery (Figure 5D). During cardiosphere culture, proliferative cells (Ki67 positive) were found primarily in the core of the sphere (Figure 5P). Many cardiospheres manifest cardiomyocyte-specific sarcomeric proteins (α MHC, cTnI) in the peripheral cells, indicative of partial myogenic differentiation (Figure 5J and 5K). On their periphery, cardiospheres also expressed CD31 (Figure 5C), CD133 (Figure 5O) and MDR-1 (Figure 5B). In their core, cardiospheres also often expressed connexin 43 (Figure 5I), Nkx2.5 (Figure 5L), and desmin (Figure 5M). Only rarely was MDR-1 expressed in the core of human cardiospheres (~5%, not shown). Cardiospheres were consistently negative for CD34 (Figure 5D) and CD45 (Figure 5N). We conclude that core cells have a cardiac progenitor immunophenotype dominated by the expression of stem cell and cardiomyocyte-related antigens. Peripheral cells represent spontaneous differentiation of precursor cells into endothelial, mesenchymal, or cardiomyogenic lineages, and/or the encapsulation of core progenitors by a subset of supportive cardiac mesenchymal cells. Thus, when CPCs are cultured directly from myocardial tissue by carefully-established methods, further sub-culture permits the formation of self-organizing cardiospheres that create a complex, niche-like environment favoring the proliferation of cardiac progenitors in their core and a surface phenotype marked by mesenchymal- and cardiac-specific antigens [19], [20].

Figure 5. Immunophenotyping of cardiospheres. Widefield images of representative cardiospheres immunostained for 16 different cardiac-related antigens.

3. The authors demonstrate increased expression of caspase-1 and IL1-beta in the OPC-CSps. It is not clear to me why this is an advantage. The authors link this activation of the inflammatory response to upregulation of stem cell markers and of growth factors but the two may result from independent effects of the OPC culture.

Author Answer: Thank you for your comments. The sub-population phenotypes of CDCs are greatly related to their cellular functions and behaviors, and the CDCs phenotypes would be adjusted according to the specific culture environment. It was also reported that the secretion cytokines of stem cells would exhibit better cytoprotective effects when they were pretreated with the cardiac injury mimicked environment [ref1]. Similarly, increasing studies explored different precondition methods to enhance cellular functions before transplantation, such as hypoxia and low concentration TNF-alpha treatment [ref2]. In addition, our previous study using pericardial fluid to simulate the in vivo inflammatory environment for CSps could improve cell viabilities.

Taking the above foundation, the OPC substrate was found to improve CSps formation in the present study, and increase expressions of caspase-1 and IL1-beta were observed in OPC-CSps. Pyroptosis is a kind of dynamic inflammation in which cells can release complex inflammatory signals to affect surrounding cells, and caspase-1 and IL1-beta are typical regulators [ref3]. Therefore, it is reasonable to assume that by using OPC substrate to induce pyroptosis of the external cells in OPC-CSps, the external pyroptotic cell could provide proper stimulus for improving the CDCs bioactivities in CSps. And the study results showed improved proliferation, mitochondria function, and oxidative stress resistance of the CDCs in OPC-CSps compared to the cells in ULA-CSps. To sum up, the present study developed the OPC substrate for CSps pretreated culture, and it could be a stable, convenient, and effective strategy to achieve mass production and cell function improvement. Still, your comment is very useful, we would perform more studies to investigate the specific regulating signaling pathway of the OPC on CDCs regulation. **To better illustrate the design and advantages of using OPC substrate to prepare CSps, we have revised the introduction section as follows:**

Introduction

Paragraph 3

Cellular physiological activities could be manipulated by the properties of the contacted substrate^{26, 27, 28}. Liquid crystal patterns could directly introduce cells into a 3D environment and form cell structures in situ²⁹, which may be beneficial for mass spheroid production during large-scale clinical applications. In addition, the alignment of monolayer support cells could form a nematic liquid crystal pattern to induce cell death at the stress localization³⁰. **Pyroptosis is an inflammation-related cell death program, and the pyroptotic cells could release complex inflammatory signals to affect surrounding cells³¹. Considering the**

inflammatory signals released from the injured cells would be more efficient in improving the cytoprotective function of the target cells than the artificial stimulus inducer¹⁵, the phenomenon of liquid crystal pattern to induce cell death might be applied to develop a local dynamic inflammatory milieu and turn this theoretical model into cell product preparation methods. Therefore, using a liquid crystal substrate for pretreated CSps might be a stable, convenient, and effective strategy to achieve mass production and cell function improvement.

15. Kshitiz. *et al.* Dynamic secretome of bone marrow-derived stromal cells reveals a cardioprotective biochemical cocktail. *Proceedings of the National Academy of Sciences* **116**, 14374-14383 (2019).
26. Kong, Y. *et al.* Regulation of stem cell fate using nanostructure-mediated physical signals. *Chem Soc Rev* **50**, 12828-12872 (2021).
27. Ji, Y.R. *et al.* Poly(allylguanidine)-Coated Surfaces Regulate TGF-beta in Glioblastoma Cells to Induce Apoptosis via NF-kappaB Pathway Activation. *ACS Appl Mater Interfaces* **13**, 59400-59410 (2021).
28. Su, N. *et al.* Membrane-Binding Adhesive Particulates Enhance the Viability and Paracrine Function of Mesenchymal Cells for Cell-Based Therapy. *Biomacromolecules* **20**, 1007-1017 (2019).
29. Kawaguchi, K., Kageyama, R., Sano, M. Topological defects control collective dynamics in neural progenitor cell cultures. *Nature* **545**, 327-331 (2017).
30. Saw, T. B. *et al.* Topological defects in epithelia govern cell death and extrusion. *Nature* **544**, 212-216 (2017).
31. Wu, Y. *et al.* Cell pyroptosis in health and inflammatory diseases. *Cell Death Discovery* **8**, 191 (2022).

References in the response

- [ref1] Kshitiz. *et al.* Dynamic secretome of bone marrow-derived stromal cells reveals a cardioprotective biochemical cocktail. *Proceedings of the National Academy of Sciences* 116, 14374-14383 (2019).
- [ref2] Zubkova, E. S. *et al.* Regulation of Adipose Tissue Stem Cells Angiogenic Potential by Tumor Necrosis Factor-Alpha. *J Cell Biochem* 117, 180-196 (2016).
- [ref3] Wu, Y. *et al.* Cell pyroptosis in health and inflammatory diseases. *Cell Death Discovery* 8, 191 (2022).

For your convenience, the related materials are listed as follows:

Ref1

BMSC Secretion in Response to Cardiac Injury Is Cytoprotective.

We next attempted to reproduce the cardioprotective effect of the BMSC conditioned media by using a reconstituted mixture of recombinant factors, matching the experimentally determined secretion profiles of BMSCs in response to injured hiPSC-CMs (Fig. 3 A and B). Strikingly, we indeed found that this mixture had a potent cytoprotective effect closely matching the effect of the BMSC-conditioned medium (Fig. 3B). To ascertain whether the combination of factors constituting the mixture increased cytoprotection to a degree greater

than any individual component, we measured the extracellular reduction of WST-8 by mitochondrial NADH, which is a correlate of cell metabolic activity and the number of viable cells (17). hiPSC-CMs were treated with hydrogen peroxide after pretreatment with the individual growth factors constituting the mixture (IGF1, HGF, and SDF-1 α) or the complete mixture, with the cell viability measured for each pretreatment. Although each component was individually cytoprotective, the application of the mixture resulted in significantly higher cellular viability of hiPSC-CMs vs. any of the individual growth factors (Fig. 3C), suggesting a combinatorial inhibition of peroxide-induced apoptosis. We used the same method to explicitly contrast the cytoprotective effects of the recombinant factors with that of various media conditioned by BMSCs, as explored in Fig. 2. In particular, we collected the medium conditioned by BMSCs, which were themselves pretreated with the media conditioned by uninjured or injured hiPSC-CMs to stimulate cardiac damage specific response. Although both these media displayed cytoprotective effects, the higher rescue from death was observed for the media obtained from BMSCs pretreated with injured cardiac cells (Fig. 3D). This rescue effect was closely matched by the effect of the mixture (Fig. 3D), indicating that the cytoprotective action of a complex, injury-specific BMSC secretome was closely approximated by a much simpler mixture composition.

We then tested the effect of the mixture in a mouse model of myocardial infarction (detailed in SI Appendix, Methods and Materials). A total of 100 μ L of mixture was injected proximally to the infarcted zone in 3 different locations after ligating the anterior coronary artery. Comparisons were made with control injections (PBS solution) and conditioned medium from BMSCs by echocardiography 3 wk after infarction. We found that the mixture injection resulted in a significant increase of cardiac output (Fig. 3E). These results suggested that the battery of factors used in our analysis was indeed characteristic of the cytoprotective effects of the conditioned medium, and that a synthetic mixture composed of the recombinant factors at levels estimated from the secretion analysis can reproduce the cytoprotective effect of BMSCs. Overall, these data indicate that the secretory signature of BMSCs is contextual to the basal or injury signals received from the cardiomyocytes, and that BMSCs can appropriately modulate their secretory response to enable tissue repair (Fig. 3F).

Fig 3. Mixture of recombinant factors mimicking secretory responses of BMSCs is cytoprotective. (A) Flow cytometry dot plots showing Annexin-V and PI staining of hiPSC-CMs treated with the recombinant mixture or media conditioned by BMSCs pretreated as indicated in Fig. 2 A–F; the recombinant mixture recapitulates the secretory signature in Fig. 2G. (B) Quantification of data shown in A quantifying cells positive for both Annexin V and PI; dye loading distributions were normal for live and dead cells for all conditions ($n = 4$ biological replicates; error bars show SEM). (C) Quantification of hiPSC-CM survival by extracellular reduction of WST-8 following treatment with 500 μ M H₂O₂ in the presence of individual factors constituting the mixture or the complete mixture ($n = 6$ biological replicates). (D) Quantification of hiPSC-CM survival by extracellular reduction of WST-8 after treatment with 500 μ M H₂O₂ following media conditioned as shown in A and B and the biochemical mixture of recombinant factors matching those in B ($n = 8$ biological replicates). (E) Cardiac output measured by echocardiography in a mouse model of myocardial infarction injected at the time of reperfusion with PBS solution (vehicle), BMSC-conditioned medium, or the recombinant mixture mimicking BMSC secretion in response to injured hiPSC-CM, along with representative image (Inset). (Error bars show SD; n.s., no significance; * $P < 0.05$, ** $P < 0.01$, and *** $P < 0.001$, Student t test.) (F) Schematic illustration showing that biological context can define the specific combinations of the factors secreted by stem cells constituting a unique soluble biochemical signature, and this signature is recognized by the target cells to trigger a desired, e.g., cytoprotective, response.

Ref2

Abstract

Tissue regeneration requires coordinated “teamwork” of growth factors, proteases, progenitor and immune cells producing inflammatory cytokines. Mesenchymal stem cells (MSC) might play a pivotal role by substituting cells or by secretion of growth factors or cytokines, and attraction of progenitor and inflammatory cells, which participate in initial stages of tissue repair. Due to obvious impact of inflammation on regeneration it seems promising to explore whether inflammatory factors could influence proangiogenic abilities of MSC. In this study we investigated effects of TNF- α on activity of adipose-derived stem cells (ADSC). We found that treatment with TNF- α enhances ADSC proliferation, F-actin microfilament assembly, increases cell motility and migration through extracellular matrix. Exposure of ADSC to TNF- α led to increased mRNA expression of proangiogenic factors (FGF-2, VEGF, IL-8, and MCP-1), inflammatory cytokines (IL-1 β , IL-6), proteases (MMPs, uPA) and adhesion molecule ICAM-1. At the protein level, VEGF, IL-8, MCP-1, and ICAM-1 production was also up-regulated. Pre-incubation of ADSC with TNF- α -enhanced adhesion of monocytes to ADSC but suppressed adherence of ADSC to endothelial cells (HUVEC). Stimulation with TNF- α triggers ROS generation and activates a number of key intracellular signaling mediators known to positively regulate angiogenesis (Akt, small GTPase Rac1, ERK1/2, and p38 MAP-kinases). Pre-treatment with TNF- α -enhanced ADSC ability to promote growth of microvessels in a fibrin gel assay and accelerate blood flow recovery, which was accompanied by increased arteriole density and reduction of necrosis in mouse hind limb ischemia model. These findings indicate that TNF- α plays a role in activation of ADSC angiogenic and regenerative potential.

Ref3

Abstract

Inflammation is a defense mechanism that can protect the host against microbe invasion. A proper inflammatory response can maintain homeostasis, but continuous inflammation can cause many chronic inflammatory diseases. To properly treat inflammatory disorders, the molecular mechanisms underlying the development of inflammation need to be fully elucidated. Pyroptosis is an inflammation-related cell death program, that is different from other types of cell death. Pyroptosis plays crucial roles in host defense against infections through the release of proinflammatory cytokines and cell lysis. Accumulating evidence indicates that pyroptosis is associated with inflammatory diseases, such as arthritis, pneumonia, and colonitis. Furthermore, pyroptosis is also closely involved in cancers that develop as a result of inflammation, such as liver cancer, esophageal cancer, pancreatic cancer, and colon cancer. Here, we review the function and mechanism of pyroptosis in inflammatory disease development and provide a comprehensive description of the potential role of pyroptosis in inflammatory diseases.

4. In figure 3, the authors measure proliferation but do not make it clear whether this is in CSps or in CDCs isolated from the CSps and seeded on the different surfaces. Subsequent experiments to measure mitochondrial function(Fig 4) and oxidative stress (Fig 5) refer to CDCs in the methods but CSps in the figure legend.

Author Answer: Thank you for your careful review. In our study, in order to examine the CDCs bioactivities in the CSps of each group, the CDCs isolated from the CSps of each group were used for the examinations of proliferation assay and mitochondrial function test. In addition, the oxidative stress resistance examination was performed with the CSps. After the CDCs were cultured on different substrate for 3 days and the CDCs have formed CSps, H₂O₂ was added to the culture medium to treat the CSps in different groups. After H₂O₂ stimulation, single cells were digested from the CSps and used for the following ROS/superoxide staining and the flow cytometry analysis. We are very sorry for ambiguous writing, and you are right about that we should make a clearer illustration. **According to your comment, we have revised the corresponding expressions in the manuscript as follows:**

Results

OPC-induced pyroptosis improved CSps cellular bioactivities and paracrine effects

.....Following a 3-day cultivation, the cell survival rate of the ULA group was markedly lower than that of the PS group. The cell survival rate of the OPC group was significantly higher than that of the ULA group ($P < 0.05$), and it showed no significant difference from the PS group (Fig. 3f). The proliferation ability of the CDCs from OPC-CSps was significantly higher than that from ULA-CSps ($P < 0.05$), and it showed no significant difference from the PS-CSps (Fig. 3g).....

Figure 3. OPC-induced pyroptosis improved CSps cellular bioactivities and paracrine effects. g Proliferation assay of the CDCs isolated from the PS-CSps, ULA-CSps, and OPC-CSps ($n = 5$).

OPC substrate improved the mitochondrial function of CSps by enhancing oxidative phosphorylation and decreasing glycolysis

...The TEM results showed that the mitochondria in the CDCs of ULA-CSps exhibited obvious swelling, vacuolization, and cristae breakage, while the mitochondria in the CDCs of OPC-CSps and the PS-CSps maintained normal cristae morphology. Furthermore, the density

of mitochondria was significantly increased in the OPC group compared to the PS group and the ULA group ($P < 0.05$) (Fig. 4i). Mitochondrial membrane potential levels of the CDCs in CSps were evaluated by immunofluorescence staining of rhodamine 123 fluorescence intensity, and the membrane potential level was significantly enhanced in the cells of OPC group compared to those of the PS group and the ULA group ($P < 0.05$) (Fig. 4j).

Figure 4. OPC substrate improved the mitochondrial function of CSps by enhancing oxidative phosphorylation and decreasing glycolysis.f Measurement of glucose uptake by CSps using 2-NBDG ($n = 3$). g Lactate release level of CSps ($n = 5$). h The ATP levels of CDCs isolated from the CSps of each group ($n = 3$). i Representative TEM images of the mitochondrial morphology from each group, and the density of the mitochondria from each group was quantified ($n = 8$ images from 2 experiments). j Rhodamine 123 staining results of mitochondrial membrane potential, and the fluorescence intensity of the CDCs isolated from CSps of each group were measured ($n = 15$ images from 3 experiments). All data are shown as the mean \pm SD, * $P < 0.05$ vs. PS, # $P < 0.05$ vs. ULA, one-way ANOVA (f-j) or two-way ANOVA (e).

OPC-CSps showed enhanced oxidative stress resistance and anti-inflammatory effect

H_2O_2 stimulation was used to test cellular oxidative stress resistance. As shown in Figure 5a and 5b, after exposure to H_2O_2 for 24 h, the fluorescence intensity of the ULA group was significantly lower than that of the PS group ($P < 0.05$). Moreover, the fluorescence intensity of the OPC group further decreased compared to the PS group and the ULA group, suggesting that OPC-CSps generated the least reactive oxygen species (ROS) and superoxide under an oxidative stress environment. Following 12 h of H_2O_2 stimulation, the percentage of viable cells in the OPC group ($73.7\% \pm 1.4\%$) was significantly higher than that in the PS group ($59.7\% \pm 7.5\%$) and the ULA group ($59.2\% \pm 5.4\%$) ($P < 0.05$) (Fig. 5c). There was no difference in the cell survival rate between the PS-CSps and the ULA-CSps. Following 24 h of H_2O_2 stimulation, a similar trend in the survival rate was observed among the three groups, and the survival rates of the PS, ULA, and OPC groups were $1.5\% \pm 0.7\%$, $5.4\% \pm 2.1\%$, and $31.7\% \pm 4.7\%$, respectively (Fig. 5d). The anti-inflammatory effects were also examined.

Following 12 h of TNF- α stimulation, the survival rates of the PS, ULA, and OPC groups were 50.0% \pm 3.7%, 53.8% \pm 4.5%, and 66.3% \pm 3.0%, which were significantly lower than the control group, while the cells from the OPC-CSps group exhibited the best anti-inflammatory effect among 3 groups (Fig. 5e).

Figure 5. Improvement in oxidative stress resistance in OPC-CSps following H₂O₂-induced injury. **a-b** Representative images and corresponding quantitative results of the (a) ROS and (b) superoxide fluorescence intensity following 24 h of H₂O₂ stimulation ($n = 15$ images from 3 experiments). **c-d** The percentages of viable cells from each group following (c) 12 h and (d) 24 h H₂O₂ stimulation were determined by Annexin V/PI flow cytometry analysis ($n = 3$). **e** The percentages of viable cells from each group following 12 h of TNF- α stimulation were determined by Annexin V/PI flow cytometry analysis ($n = 3$). All data are shown as the mean \pm SD, * $P < 0.05$ vs. Control, # $P < 0.05$ vs. PS, & $P < 0.05$ vs. ULA, one-way ANOVA.

Methods and Materials

Metabolic analysis of CSps

The glucose uptake ability of CDCs was evaluated by using the fluorescent glucose 2-NBDG (Cayman). After 3 days of cultivation of CDCs on different substrates, all the culture medium

was removed and replaced with glucose-free DMEM containing 50 μ M 2-NBDG for 30 min. Consequently, the fluorescence intensity of the cells was measured by flow cytometry (BD FACSCanto). Following a 3-day cultivation, the culture medium of each group was collected to test the extracellular lactate contents, and the reagent kit was the Lactate Colorimetric Assay Kit (Nanjing JianCheng).

For the intracellular ATP content assay, CDCs were cultured on different substrates for 3 days and formed CSps, and single cells were isolated from the CSps of each group and seeded in a 96-well plate at a density of 1×10^5 . The intracellular ATP content was determined by using the CellTiter-Glo Luminescent Cell Viability Assay (Promega). For the mitochondrial membrane potential measurement, digested cells were washed twice with PBS and incubated with 2 μ M Rho123 (Beyotime) for 20 min in a dark environment at 37 °C. Fluorescence images were taken under a fluorescence microscope (Olympus, FV3000), and the fluorescence intensity of the cells was analyzed with ImageJ software.

Oxidative stress resistance level measurement

Following a 3-day cultivation, CDCs formed CSps on different culture substrate, and the CSps were subjected to H₂O₂ stimulation for 24 h. And then single cells were isolated from the CSps and used for intracellular ROS and superoxide production measurement with the cellular ROS/superoxide detection assay kit (Abcam). In short, the isolated cells were incubated for ROS/superoxide detection for 30 min at 37 °C. The results were observed and recorded by a laser confocal scanning microscope (Olympus, FV3000). In addition, after the CSps were exposed following 12 h or 24 h of exposure to H₂O₂, CSps were digested into single cells and used to measure the cell survival rate with the Alexa Fluor® 488 Annexin-V/Dead Cell Apoptosis Kit (Invitrogen). The results were analyzed by flow cytometry (BD FACSCanto).

Anti-inflammation ability measurement

Following a 3-day cultivation, CDCs formed CSps on different culture substrate, and the CSps were subjected to 50 ng/ml TNF- α stimulation for 12 h. And then single cells were isolated from the CSps and used to measure the cell survival rate with the Alexa Fluor® 488 Annexin-V/Dead Cell Apoptosis Kit (Invitrogen). The results were analyzed by flow cytometry (BD FACSCanto).

5. The results section of the RNA-seq data is confusing. The authors state that “the DEGs between the OPC-CSps and the ULA-CSps were enriched in the glycolytic/glycogenic pathway and the HIF-1 signaling pathway” implying that these pathways were higher in the OPC-CSps than the ULA-CSps. But then they say “Moreover, gene set enrichment analysis (GSEA) also revealed the downregulation of the hypoxic and glycolytic components in the OPC-CSps compared to the ULA-CSps”. They go on to show that glycolysis is lower in the OPC-CSps (or CDCs) than the other two groups which is interesting as normally

stemness is associated with glycolysis and the authors have shown increased proliferation (associated with stemness) and stem cell genes in this population.

Author Answer: Thank you for your comments. We are sorry for the poor expression that led to the misunderstanding, but we're afraid that we can't agree on your opinion. (1) Firstly, the Kyoto Encyclopedia of Genes and Genomes (KEGG) enrichment analysis could show the most significant pathways between two groups according to their DEGs, and the ranking result of the KEGG enrichment analysis cannot interpret the expression levels between the compare groups, and that is the reason why we further performed the GSEA analysis.

For the interpretation of the GSEA analysis, a group with a higher absolute value of the biggest enrichment score (ES) represents it has a higher expression level on the genes in that pathway. In the comparison between the OPC-CSps and the ULA-CSps on glycolysis (as shown in the below figure), the absolute value of the biggest ES of ULA-CSps group was 0.48, and that was 0.11 of the OPC-CSps group, so it means the expressions of the glycolysis-related genes are higher in the ULA-CSps than the OPC-CSps. By this mean, we could conclude that the glycolysis levels among three groups are ULA>OPC>PS. Also, the results of the mRNA transcription levels of the genes in the glycolytic pathway (HK2, LDHA, and PFKL) also showed that the glycolysis were higher in the ULA-CSps than the OPC-CSps (Fig4e).

Secondly, the stem cell metabolism emerges as a critical determinant of cellular processes and is uniquely adapted to support proliferation, stemness, and commitment. During the quiescent states, the stem cells mainly generate energy through glycolysis while maintaining low oxidative phosphorylation (OxPhos), providing metabolites for biosynthesis of macromolecules. And when the stem cells receive injury signals, there would be a shift in cellular metabolism towards OxPhos and enhance proliferation, and cellular functions [Ref 1]. For stem cell therapy in MI, one major problem was that the transplanted cells would face a harsh microenvironment, and precondition treatments to simulate the MI microenvironment have been reported to enhance cellular functions in vivo. In the present study, taking the idea of using the OPC substrate to provide proper stimulus for the CSps to enhance cellular functions before transplantation, we found the cells in OPC-CSps exhibited better proliferation, paracrine effects, and oxidative stress resistance with enhanced OxPhos levels

compare to the ULA-CSps, which is consistent to the change of stem cell metabolism during cardiac injury and showed the benefits of using OPC substrate for CSps culture. **The revisions for a clearer result illustration are as follows:**

Results

OPC substrate improved the mitochondrial function of CSps by enhancing oxidative phosphorylation and decreasing glycolysis

The results of RNA-sequencing (RNA-Seq) analysis showed that there were 1251 differentially expressed genes (DEGs) between the ULA group and the OPC group (Fig. 4a), and 232 DEGs were not among the DEGs of PS vs. ULA or PS vs. OPC (Fig. 4b). Kyoto Encyclopedia of Genes and Genomes (KEGG) enrichment analysis showed that **metabolic pathways, glycolytic/glycogenic pathway, and the HIF-1 signaling pathway are the top 3 significant signaling pathways of the DEGs between the OPC-CSps and the ULA-CSps** (Fig. 4c). Moreover, gene set enrichment analysis (GSEA) also revealed the downregulation of the hypoxic and glycolytic components in the OPC-CSps compared to the ULA-CSps (Fig. 4d). The oxidative phosphorylation genes in the OPC group, including *CS*, *COXII*, *IDH2*, *SDHA*, and *MDH2*, were notably upregulated compared with those in the PS and ULA groups ($P < 0.05$). Compared to the ULA group, the key genes of the glycolytic pathway, *HK2*, *LDHA*, and *PFKL*, dramatically decreased in the OPC group ($P < 0.05$) (Fig. 4e).

Figure 4. OPC substrate improved the mitochondrial function of CSps by enhancing oxidative phosphorylation and decreasing glycolysis.

a Hierarchical cluster analysis of upregulated (red) and downregulated (blue) genes after culture on different substrates for three days. **b** The DEGs from the hierarchical cluster analysis were interpreted in the Venn diagram. The DEGs with a $|\log_2FC| \geq 1$ and an adjusted P value ≤ 0.05 were identified by DESeq (1.28.0). **c** KEGG analysis of the top 7 significant pathways (P value < 0.05).

[Ref 1] Vagner O. C. Rigaud, et al. Stem Cell Metabolism: Powering Cell-Based Therapeutics. Cells. 9, 2490 (2022).

For your convenience, the related materials are listed as follows:

Ref 1

2.1. Quiescence

Adult stem cells are tissue resident stem cells thought to support tissue homeostasis and cellular turnover following injury [23] (Wagers and Weissman 2004) [23]. In uninjured tissues, these cells live in a “latent” cell cycle-arrested state known as quiescence. Although quiescent cells do not divide, they retain the ability to re-enter the cell cycle and proliferate in response to environmental stimuli [6,24]. Most of the adult stem cells, including hematopoietic stem cells (HSCs) and cardiac stem/progenitor cells, are found to reside in hypoxic niches in a quiescent or slow-cycling stage [7,25–27]. This low-oxygen microenvironment is not only tolerated by these cells, but also seems to be essential for their function. While poorly understood, survival in hypoxic niches requires significant metabolic adaptations with the quiescent stem cell mainly operating under glycolysis that shifts to OXPHOS once cells proliferate or commit towards cardiac lineages (Figure 1).

Figure 1. Stem cell metabolism is dynamically modulated to control stemness, proliferation, and cell commitment. Quiescent stem cells are mostly glycolytic due to HIF-1 activity in the hypoxic niche with low generation of ROS to maintain stemness. Outside the hypoxic niche, the oxygen levels begin to rise increasing the oxidative phosphorylation (OxPhos) and reactive oxygen species (ROS) levels, which stimulate the cells to proliferate and differentiate. During proliferation, stem cells mainly rely on glycolysis while still maintaining low OxPhos levels to fuel the cells with biosynthetic intermediates important for cell growth. Stem cell differentiation to cardiomyocytes, however, depends on a metabolic shift from glycolysis to OxPhos in a ROS-dependent manner.

2.2. Proliferation

Quiescent adult stem cells are reversibly arrested in G₀ to maintain their stemness [5]. However, following stimulation or injury, quiescent cells quickly re-enter the cell cycle, proliferate, and commit into specific tissue lineages to replace damaged cells [29]. The transition from a quiescent to a lineage-committed state is characterized by migration from a hypoxic niche to an oxygen-rich microenvironment [5]. In the presence of oxygen, mitochondrial activity increases, which generates ROS, believed to induce adult stem cell proliferation but also apoptosis at elevated concentrations [24,29]. Thus, energy metabolism in proliferating cells markedly differs from that in quiescent cells.

With higher oxygen tension, HIF-1 levels decrease through oxygen-mediated ubiquitination and proteasomal degradation, which affects proliferation [5]. Levels of HIF-1 target genes, such as PDK2 and PDK4, decrease, dephosphorylating PDH and leading to the oxidation of pyruvate into acetyl-CoA [5]. Increased acetyl-CoA feeds the TCA cycle, increasing mitochondrial respiration and thereby switching the metabolism from anaerobic glycolysis to OxPhos. Indeed, mesenchymal stem cells (MSCs) cultured under normoxia have been shown to rely on OxPhos and high oxygen consumption [36]. Additionally, these cells present an increased proliferation rate [37]. However, the metabolic switch to OxPhos is associated with higher mitochondrial activity, which generates ROS and over a long time leads to a significant increase in senescence, which impairs MSC stemness [37].

6. In the in vivo experiment, the authors again don't make it clear whether they inject CSps or CDCs. They refer to OPC-CSps Matrigel suspensions but then give a cell number which would be hard to determine from a sphere culture. In the ex vivo histology, they show a measurement of the 'infarct wall' in the sham, which has no infarct and the staining in Fig 7b suggests there are no major blood vessels in the sham heart, which is unlikely.

Author Answer: Thank you for your comments.

(1) It is an important question to ensure the transplanted cell number. In fact, after we collect the OPC-CSps from the culture substrate, we would take the CSps sample and perform a DNA test, and a standard curve of cell number and DNA content would be established and used to calculate the correspond cell number of the CSps sample. By this mean, we can quantify the transplanting cell number of the OPC-CSps.

(2) As for the second question, it is undoubtable that there are major blood vessels in the heart tissue of the animals in the sham group, which has also been shown in the Masson staining result of the sham heart, and you are right we should display an image that could be representative for the blood vessel status of the healthy myocardium. In addition, the present study used the left anterior descending coronary artery (LAD) ligation model, so we showed the left ventricle area to display the infarcted myocardium. According to your comment, we have revised the figure legend "infarct wall" to "left ventricle". **Thank you again for your valuable comments, they improve the quality of our manuscript a lot, and the corresponding revisions are as follows:**

Figure 7. The reduction in cardiac inflammation, apoptosis, and hypertrophy after OPC-CSps transplantation. **a** Representative images of Masson trichrome staining 12 weeks following MI, and yellow arrows mark the blood vessels in the border of the infarct area. **b** Representative images of CD68⁺ macrophages in the peri-infarct zone 2 weeks following MI, α -SMA⁺ CD31⁺ vessels in the infarct zone 12 weeks following MI, and TUNEL⁺ cells in the peri-infarct zone 12 weeks following MI. Spilt channels of the α -SMA and CD31 staining results are shown in Supplementary Fig. 5. **c** Quantitative results of the infarct area ($n = 8$ rats). **d** The percentage of viable myocardium at the infarct area ($n = 8$ rats). **e** Quantitative results of infarct wall thickness ($n = 8$ rats). **f** The quantitative results of CD68⁺ macrophages per HPF assessed by ImageJ software. HPF, high-power field ($n = 15$ images from 3 rats). **g** The quantitative results of vessel density of each group ($n = 15$ images from 8 rats). **h** The quantitative results of the TUNEL⁺ rate assessed by ImageJ software ($n = 15$ images from 8 rats). **i** Representative images of WGA staining of heart tissues shown in different regions at 12 weeks. The cardiomyocyte membrane was stained with WGA (green), cardiomyocytes were identified by staining for cTnT (red), and

DAPI showed nuclei. **j-l** Quantitative analysis of cardiomyocyte cross-sectional area from **(j)** the left ventricle, **(k)** the border zone, and **(l)** the remote zone ($n = 15$ images from 8 rats). Each data point is represented as the mean \pm SD, $*P < 0.05$ vs. sham, $\#P < 0.05$ vs. vehicle, Student's t test (c&d) or one-way ANOVA (e, f, g, h, j, k, and l).

Supplementary Fig. 5. Spilt channels images of the α -SMA and CD31 immunofluorescence staining results of Fig. 7b. Paraffin sections of the hearts from each group were used for analysis. The cell nucleus was detected by staining with Dapi (blue), and the expression of anti-CD31 (endothelial cell marker) antibodies (green) and anti- α -SMA (smooth muscle cell marker) antibodies (red) were co-stain to determine the vascular distribution of the left ventricle area.

7. In the discussion on page 25, the authors refer to microenvironmental cues from ECM increasing expression of CD31 and CD34 but there is no discussion of ECM in the various surfaces. The discussion of the mitochondrial changes shows confused interpretation of these data. The authors say “In this study, the oxidative phosphorylation of OPC-CSps was enhanced, while the ULA-CSps altered their energy production toward glycolysis” and then they say “In addition, the OPC group showed a significant decrease in glucose uptake levels (Fig. 4f), suggesting that the OPC-CSps may survive longer in nutrient-limited conditions”. If oxidative phosphorylation is enhanced then the cells still require glucose, since low levels of other substrates are provided in the cell culture medium used. Furthermore, if cells are injected into the infarct zone, where oxygen supply is limited, then they will need to rely on glycolysis.

Author Answer: Thank you for your insightful comments.

Firstly, it is a good idea to replenish the discussion about the increasing expression of CD31 and CD34. CD31 and CD34 are the typical markers for the endothelial sub-population cells of the CDCs. In this study, according to the KEGG enrichment analysis, the focal adhesion pathway are the most significant pathways between the PS group and the ULA group, and the PS group and the OPC group. The DEGs of the focal adhesion pathway indicated great influence on the ECM components related genes expressions, such as Lama2, Itgb1, Lamc1, Col6a3, Col2a1. As the result showed, 3D CSps are more efficiently formed in the ULA group and the OPC group than the PS group. It is widely studied that the ECM components and the cell-cell interactions within the 3D cells spheroids would have significant changes than the 2D culture. Taken together, CDCs in 3D spheroids may enhance their phenotypes towards endothelial cells, which is an interesting question. But still, more studies are needed to carry out to find out the specific regulating mechanism, and we will keep on focusing on this question in our future studies.

Secondly, it is a good question to consider the nutrition supply for the transplanted cells in the pericardial cavity in vivo. The pericardial cavity is a double-walled sac, and the pericardial fluid in the pericardial cavity was studied as plasma ultrafiltrate with specific characteristics just like the pleura fluid, so there would be oxygen and nutrition supply (such as glucose) for the transplanted cells. In addition, there are local cells in the pericardial fluid which could prove the nutrition supply is adequate for the cells in the cavity [Ref 1].

According to your comment, we have revised the discussion section as follows:

Discussion, paragraph 5

In addition, cell–cell and cell-ECM contacts also greatly affect the phenotypes of the cells in CSps. Compared to the PS group, the expression of CD31 and CD34 significantly increased in the ULA group and the OPC group (Fig. 2d), and the focal adhesion pathway of these 2 groups significantly differed from the PS group, which could be related to their higher efficiency in 3D spheroid formation. It is widely studied that the ECM components and the cell-cell contacts within the 3D cells spheroids have significant changes than the 2D culture, and these results showed CDCs in 3D spheroids may enhance their phenotypes towards endothelial cells via the change of microenvironmental cues in the ECM³⁶, but more studies are needed to verify this assumption. The different portions of CD90⁺ cells could be observed when cultured on the PS, ULA, and OPC substrates, so it is reasonable to assume that the expression level of CD90 could be regulated by the physical and chemical environments provided by each substrate. In addition, several studies have shown that a decrease in mesenchymal markers, such as CD90, could lead to an improvement in the pluripotency of spheroid cells^{37, 38}. In this study, compared with the PS group and the ULA group, the OPC group showed the lowest expression level of CD90 and the highest expression levels of pluripotency markers, including *Nanog*, *Sox2*, and *Oct4* (Fig. 3h).

[Ref 1] Vogiatzidis, K. et al. Physiology of pericardial fluid production and drainage. Front. Physiol. 6, 62 (2015).

For your convenience, the related materials are listed as follows:

Ref 1

Pericardial fluid

The composition of the normal human pericardial fluid is difficult to define. All available data have been obtained either by cardiothoracic surgery patients or from animals. This probably compromises the data validity (Ben-Horin et al., 2005). However, the pericardial fluid is a plasma ultrafiltrate having specific characteristics just like the pleura fluid (Mauer et al., 1940; Holt, 1970). Volumetric studies have shown that the pericardial fluid volume is directly analogous to the animal size: in rabbits 0.4–1.9 mL, in dogs 0.5–2.5 mL and in adult humans about 20–60 mL (average 15–35 mL) (Vesely and Cahill, 1986; Ben-Horin et al., 2005).

Pericardial fluid coloring studies report that the fluid distribution inside the cavity is heterogeneous. The largest amount is inside the atrioventricular and the intraventricular sulcus, the superior and the transversal sinus, especially on the supine position (D'Avila, 2003). Nevertheless, there are some pharmacokinetic studies that show that the pericardial fluid is stirring up constantly and thus the supplement's composition is the same regardless the position (Chinchoy, 2005).

Regarding the cell population, studies in human normal pericardial fluid have shown the presence of a heterogenous cell population. There are mesothelial cells, lymphocytes (53%), glanulocytes (31%), macrophages (12%), eosinophils (1.7%), and basophils (1.2%). This means that the pericardial fluid “lymphocytosis” should always be under critical consideration and characterized as pathological only when it exceeds 60% of the whole cell population (Gibson and Segal, 1978a; Benhaiem-Sigaux et al., 1985).

Response to reviewer 2:

1. The authors mention (see line 127) fibroblast-like supporting cells with CSps but these cells are not visible in the Figure 2. Please include images with high magnification to appreciate these cells. What is the percentage of the fibroblast-like supporting cells? I suppose that these cells are included in the phenotypic analyses. In my opinion, the percentage of these cells would allow better interpretation of the results and should therefore be included in the manuscript.

Author Answer: Thank you for your comments. Firstly, we are very sorry for the expressions in the manuscript. After we went through the references, the fibroblast-like cells around CSps would be more precise to be described as the mesenchymal-like cells around the CSps. CSps are isolated and cultured from the heart tissues and they contained a mix population of cells expressed related markers, including cardiac progenitor cells (CPCs), endothelial cells, and mesenchymal cells, and CD90 and CD105 was typical mesenchymal cell markers of the CDCs [ref1]. You are right about that we should identify the sub-cell types of the CSps, so we performed the flow cytometry analysis to verify the portions of different cell population of the CSps in the present study. According to your comment, we have revised the writing in the manuscript for better illustration, and we have also replenished high magnification fields in the supplementary materials to better show the formation process of the CSps on different substrates, where the morphology of the surrounding cells of PS-CSps could be clearly seen. In addition, the percentage of different markers are replenished in the manuscript as you suggested. **The corresponding revisions are as follows:**

The OPC substrate promoted CSps formation and progenitor phenotypes

...The expression levels of CSps surface markers were analyzed. Compared to the PS group (KDR: 3.15±0.85 %, Sca-1: 12.61±3.63 %) and the ULA group (KDR: 3.76±0.79 %, Sca-1: 11.13±2.39 %), the OPC group exhibited the highest expression of KDR (8.55±1.20 %) and Sca-1 (26.17±4.17%) ($P<0.05$). In addition, the ULA group (CD31: 11.74±0.89 %, CD34: 13.91±1.64 %) and the OPC group (CD31: 11.99±1.07 %, CD34: 14.47±0.85 %) showed an increase in the expression of CD31 and CD34 when compared to the PS group (CD31: 2.68±1.24 %, CD34: 1.80±0.55 %) ($P<0.05$), along with a decrease in CD90 (PS: 98.47±0.65 %, ULA: 68.37±4.81 %, OPC: 6.87±1.01 %) ($P<0.05$) and no significant difference in CD105 compared to the PS group (PS: 45.73±7.03 %, ULA: 37.03±3.50 %, OPC: 39.53±8.51 %) (Fig. 2d).

2. Please include images with high magnification to appreciate the morphological differences of CSps cultured on PS, ULA and OPC.

Author Answer: Thank you for your comments. It is a good idea. We also considered this problem when selecting pictures for display. Low magnification fields can intuitively show the pellet-forming efficiency of CSps in different petri dishes, while high magnification fields can clearly see the structure of CSps and the morphology of surrounding cells. According to your suggestion, we have added the images in high magnification fields in the supplementary materials, and **the corresponding revisions are as follows:**

Results

The OPC substrate promoted CSps formation and progenitor phenotypes

The formation of CSps on the polystyrene (PS) substrate, the ultralow attachment (ULA) substrate and the OPC substrate was observed, and different shapes of CSps were acquired (Fig. 2a and Supplementary Fig.1). The adherent CDCs on the PS substrate converged and formed regular **spherical aggregates**, and supporting cells **surrounded at the base** of the PS-CSps. The ULA-CSps were formed by suspended single-cell stacking, and they developed into noncircular, oval, and irregular shapes. On the OPC substrate, CDCs initially attached to the substrate and gradually aggregated to form regular and circular spheroids, and no supporting cells were observed around the OPC-CSps. During the formation of CSps, the **sizes of OPC-CSps** showed a stable **growth** tendency compared with **the obvious increase** of the PS-CSps and the ULA-CSps (Fig. 2b and Supplementary Fig.1). Following a 3-day cultivation, the sizes of PS-CSps, ULA-CSps, and OPC-CSps were $280.96 \pm 40.56 \mu\text{m}$, $203.34 \pm 69.36 \mu\text{m}$, and $120.29 \pm 15.34 \mu\text{m}$, respectively. In addition, the spheroid density on the OPC substrate was significantly higher than that on the other two substrates (Fig. 2c).

Figure S1. Morphological changes in CSps on the PS, ULA, and OPC substrates in high magnification fields.

3. How do the authors explain that the size of OPC-CSps remains stable during culture even though the cells proliferate more and the survival rate is high with OPC.

Author Answer: Thank you for your comments. This is a good question. Firstly, the size and the 3D structure of CSps are greatly related to the characteristics of the culture substrate. On the ULA substrate, cells could hardly adhesive to the culture dish, so the cells highly aggregated and formed 3D cell spheroid. Because of the large size of the ULA-CSps, so the proliferated cells in the ULA-CSps mainly distributed in the periphery of the CSps where the cells could get adequate nutrition supply. On the OPC substrate, the cells were induced by unique mechanical force and formed CSps, in which the cells remained certain cell-cell spaces for cells to proliferate and allowing nutrition transportation, so the size of the OPC-CSps did not exhibit significant changes during the 3-day culture period. **To support this illustration, we have replenished the TEM ultrastructure analysis to show the cell-cell structure in different CSps.**

It was reported that the sub-population phenotypes of CDCs are greatly related to their cellular functions and behaviors, and the CDCs phenotypes would be adjusted according to the specific culture environment. It was also reported that the secretion cytokines of stem cells would exhibit better cytoprotective effects when the they were pretreated with the cardiac injury mimicked environment [ref1]. Similarly, increasing studies explored different precondition methods to enhance cellular functions before transplantation, such as hypoxia and low concentration TNF-alpha treatment [ref2]. In the present study, the proliferation assay was performed with the isolated cells (CDCs) from CSps in different groups. Following the culture on OPC-substrates, the inner cells in the OPC-CSps would receive the inflammatory signals from the periphery pyroptotic cells, and enhanced cellular metabolisms were observed in the results, and these are the important reasons for the improved proliferation and cell survival. **The corresponding revisions are as follows:**

OPC-induced pyroptosis improved CSps cellular bioactivities and paracrine effects

...The transmission electron microscope (TEM) analysis was performed to evaluate the cellular ultrastructure. Highly cell aggregation was observed in the ULA-CSps, while there was certain cell-cell space remained in the center of the OPC-CSps (Supplementary Fig. 2). Normal structures of the nucleus, mitochondria, and rough endoplasmic reticulum were observed in the PS group. However, endoplasmic reticulum dilatation, degranulation, and swollen mitochondria were observed in the cells from the ULA group. Moreover, cells in the OPC-CSps showed normal nuclear structures and abundant normal mitochondria. In contrast to the PS group and the ULA group, many microvesicles could be observed on the membrane surface (Fig. 3e)....

Figure 3. OPC-induced pyroptosis improved CSps cellular bioactivities and paracrine effects..... e Representative TEM images of the cell ultrastructure in CSps. N: nucleus, yellow arrows indicate mitochondria, yellow asterisks (*) indicate endoplasmic reticulum, and yellow triangles (Δ) indicate microvesicles.

Supplementary Fig. 2. TEM analysis results of the cellular ultrastructure in different groups. The ultrastructure of CDCs from the PS group, the ULA group, and the OPC group are shown. Highly cell aggregation was observed in the ULA-CSps, while the cells in OPC-CSps remained certain cell-cell spaces and normal cell ultrastructure.

4. The authors mention that the expression of caspase-1 was observed in the peripheral cells of the OPC-CSps. Thus, proliferative cells are found in the center of CSps? Immunostaining images showing cell proliferation should be included in the manuscript to better appreciate these results.

Author Answer: Thank you for your comments. It is a good suggestion to improve the quality of our manuscript, and we have replenished an immunofluorescence staining assay to valid the expression of proliferation marker Ki67 in the CSps, and the results showed that the proliferated cells present in the center of the OPC-CSps and PS-CSps, while the Ki67 positive cells were mostly on the periphery of the ULA-CSps. **The corresponding revisions are showed as follows:**

OPC-induced pyroptosis improved CSps cellular bioactivities and paracrine effects

...The expression of Ki67 was observed in the center of the PS-CSps and the OPC-CSps, while the Ki67⁺ cells were mostly found in the periphery of the ULA-CSps (Supplementary Fig. 3). Additionally, the proliferation ability of the CDCs from OPC-CSps was significantly higher in the OPC-CSps than that from ULA-CSps ($P<0.05$), and it showed no significant difference from the PS-CSps (Fig. 3g)....

Supplementary Fig. 3. Representative images of Ki67 immunofluorescence staining results of the CSps from each group following 3 days of cultivation.

References in the response

[ref1] Kshitiz. et al. Dynamic secretome of bone marrow-derived stromal cells reveals a cardioprotective biochemical cocktail. Proceedings of the National Academy of Sciences 116, 14374-14383 (2019).

[ref2] Zubkova, E. S. et al. Regulation of Adipose Tissue Stem Cells Angiogenic Potential by Tumor Necrosis Factor-Alpha. J Cell Biochem 117, 180-196 (2016).

For your convenience, the related materials are listed as follows:

Ref1

BMSC Secretion in Response to Cardiac Injury Is Cytoprotective.

We next attempted to reproduce the cardioprotective effect of the BMSC conditioned media by using a reconstituted mixture of recombinant factors, matching the experimentally determined secretion profiles of BMSCs in response to injured hiPSC-CMs (Fig. 3 A and B). Strikingly, we indeed found that this mixture had a potent cytoprotective effect closely matching the effect of the BMSC-conditioned medium (Fig. 3B). To ascertain whether the combination of factors constituting the mixture increased cytoprotection to a degree greater than any individual component, we measured the extracellular reduction of WST-8 by mitochondrial NADH, which is a correlate of cell metabolic activity and the number of viable cells (17). hiPSC-CMs were treated with hydrogen peroxide after pretreatment with the individual growth factors constituting the mixture (IGF1, HGF, and SDF-1 α) or the complete mixture, with the cell viability measured for each pretreatment. Although each component was individually cytoprotective, the application of the mixture resulted in significantly higher cellular viability of hiPSC-CMs vs. any of the individual growth factors (Fig. 3C), suggesting a combinatorial inhibition of peroxide-induced apoptosis. We used the same method to explicitly contrast the cytoprotective effects of the recombinant factors with that of various media conditioned by BMSCs, as explored in Fig. 2. In particular, we collected the medium conditioned by BMSCs, which were themselves pretreated with the media conditioned by uninjured or injured hiPSC-CMs to stimulate cardiac damage specific response. Although both these media displayed cytoprotective effects, the higher rescue from death was observed for the media obtained from BMSCs pretreated with injured cardiac cells (Fig. 3D). This rescue effect was closely matched by the effect of the mixture (Fig. 3D), indicating that the cytoprotective action of a complex, injury-specific BMSC secretome was closely approximated by a much simpler mixture composition.

We then tested the effect of the mixture in a mouse model of myocardial infarction (detailed in SI Appendix, Methods and Materials). A total of 100 μ L of mixture was injected proximally to the infarcted zone in 3 different locations after ligating the anterior coronary artery. Comparisons were made with control injections (PBS solution) and conditioned medium from BMSCs by echocardiography 3 wk after infarction. We found that the mixture injection resulted in a significant increase of cardiac output (Fig. 3E). These results suggested that the battery of factors used in our analysis was indeed characteristic of the cytoprotective effects of the conditioned medium, and that a synthetic mixture composed of the recombinant factors at levels estimated from the secretion analysis can reproduce the cytoprotective effect of BMSCs. Overall, these data indicate that the secretory signature of BMSCs is contextual to the basal or injury signals received from the cardiomyocytes, and that BMSCs can appropriately modulate their secretory response to enable tissue repair (Fig. 3F).

Fig 3. Mixture of recombinant factors mimicking secretory responses of BMSCs is cytoprotective. (A) Flow cytometry dot plots showing Annexin-V and PI staining of hiPSC-CMs treated with the recombinant mixture or media conditioned by BMSCs pretreated as indicated in Fig. 2 A–F; the recombinant mixture recapitulates the secretory signature in Fig. 2G. (B) Quantification of data shown in A quantifying cells positive for both Annexin V and PI; dye loading distributions were normal for live and dead cells for all conditions ($n = 4$ biological replicates; error bars show SEM). (C) Quantification of hiPSC-CM survival by extracellular reduction of WST-8 following treatment with 500 μ M H₂O₂ in the presence of individual factors constituting the mixture or the complete mixture ($n = 6$ biological replicates). (D) Quantification of hiPSC-CM survival by extracellular reduction of WST-8 after treatment with 500 μ M H₂O₂ following media conditioned as shown in A and B and the biochemical mixture of recombinant factors matching those in B ($n = 8$ biological replicates). (E) Cardiac output measured by echocardiography in a mouse model of myocardial infarction injected at the time of reperfusion with PBS solution (vehicle), BMSC-conditioned medium, or the recombinant mixture mimicking BMSC secretion in response to injured hiPSC-CM, along with representative image (Inset). (Error bars show SD; n.s., no significance; * $P < 0.05$, ** $P < 0.01$, and *** $P < 0.001$, Student t test.) (F) Schematic illustration showing that biological context can define the specific combinations of the factors secreted by stem cells

constituting a unique soluble biochemical signature, and this signature is recognized by the target cells to trigger a desired, e.g., cytoprotective, response.

Ref2

Abstract

Tissue regeneration requires coordinated “teamwork” of growth factors, proteases, progenitor and immune cells producing inflammatory cytokines. Mesenchymal stem cells (MSC) might play a pivotal role by substituting cells or by secretion of growth factors or cytokines, and attraction of progenitor and inflammatory cells, which participate in initial stages of tissue repair. Due to obvious impact of inflammation on regeneration it seems promising to explore whether inflammatory factors could influence proangiogenic abilities of MSC. In this study we investigated effects of TNF- α on activity of adipose-derived stem cells (ADSC). We found that treatment with TNF- α enhances ADSC proliferation, F-actin microfilament assembly, increases cell motility and migration through extracellular matrix. Exposure of ADSC to TNF- α led to increased mRNA expression of proangiogenic factors (FGF-2, VEGF, IL-8, and MCP-1), inflammatory cytokines (IL-1 β , IL-6), proteases (MMPs, uPA) and adhesion molecule ICAM-1. At the protein level, VEGF, IL-8, MCP-1, and ICAM-1 production was also up-regulated. Pre-incubation of ADSC with TNF- α -enhanced adhesion of monocytes to ADSC but suppressed adherence of ADSC to endothelial cells (HUVEC). Stimulation with TNF- α triggers ROS generation and activates a number of key intracellular signaling mediators known to positively regulate angiogenesis (Akt, small GTPase Rac1, ERK1/2, and p38 MAP-kinases). Pre-treatment with TNF- α -enhanced ADSC ability to promote growth of microvessels in a fibrin gel assay and accelerate blood flow recovery, which was accompanied by increased arteriole density and reduction of necrosis in mouse hind limb ischemia model. These findings indicate that TNF- α plays a role in activation of ADSC angiogenic and regenerative potential.

5. Does H₂O₂ treatment increase the level of caspase-1 and Il-1 β level at the peripheral cells?

Author Answer: Thank you for your comments. H₂O₂ is a major member of ROS, and it is also a critical upstream signaling molecule in regulating pyroptosis, so the H₂O₂ treatment would cause the increasing expression of caspase-1 and Il-1 β . And that is the experimental principal of the anti-oxidative assay. However, it is worthwhile to aware that H₂O₂ is a small molecular, it would not only have effect on the periphery cells of CSps, but it would also permeate into the center of the CSps and lead to a wide range of pyroptosis. Compared to the present study, the CSps was formed under the mechanical induction of OPC substrate, and only the periphery cells were induced pyroptotic, while the cells in the center of the OPC-CSps received the stimulation signals from the periphery cells and enhanced cell function.

6. Figure 5 needs to be revised. The images should illustrate CSps not separated cells. Idem for Figure 4j.

Author Answer: Thank you for your careful review. In our study, in order to examine the CDCs bioactivities in the CSps of each group, the CDCs isolated from the CSps of each group were used for the examinations of proliferation assay and mitochondrial function test (Fig 4h&j). In addition, the oxidative stress resistance examination (Fig 5) was performed with the CSps. After the CDCs were cultured on different substrate for 3 days and the CDCs have formed CSps, H₂O₂ was added to the culture medium to treat the CSps in different groups. After H₂O₂ stimulation, single cells were digested from the CSps and used for the following ROS/superoxide staining and the flow cytometry analysis. We are very sorry for ambiguous writing, and you are right about that we should make a clearer illustration. **According to your comment, we have revised the corresponding expressions in the manuscript as follows:**

Results

OPC-induced pyroptosis improved CSps cellular bioactivities and paracrine effects

...Following a 3-day cultivation, the cell survival rate of the ULA group was markedly lower than that of the PS group. The cell survival rate of the OPC group was significantly higher than that of the ULA group ($P < 0.05$), and it showed no significant difference from the PS group (Fig. 3f). The expression of Ki67 was observed in the center of the PS-CSps and the OPC-CSps, while the Ki67⁺ cells were mostly found in the periphery of the ULA-CSps (Supplementary Fig. 3). Additionally, the proliferation ability of the CDCs from OPC-CSps was significantly higher in the OPC-CSps than that from ULA-CSps ($P < 0.05$), and it showed no significant difference from the PS-CSps (Fig. 3g)....

Figure 3. OPC-induced pyroptosis improved CSps cellular bioactivities and paracrine effects. g Proliferation assay of the CDCs isolated from the PS-CSps, ULA-CSps, and OPC-CSps ($n = 5$).

OPC substrate improved the mitochondrial function of CSps by enhancing oxidative phosphorylation and decreasing glycolysis

The TEM results showed that the mitochondria in the CDCs of ULA-CSps exhibited obvious swelling, vacuolization, and cristae breakage, while the mitochondria in the CDCs of OPC-CSps and the PS-CSps maintained normal cristae morphology. Furthermore, the density of mitochondria was significantly increased in the OPC group compared to the PS group and the ULA group ($P < 0.05$) (Fig. 4i). Mitochondrial membrane potential levels of the CDCs in CSps

were evaluated by immunofluorescence staining of rhodamine 123 fluorescence intensity, and the membrane potential level was significantly enhanced in the cells of OPC group compared to those of the PS group and the ULA group ($P < 0.05$) (Fig. 4j).

Figure 4. OPC-CSps exhibited enhanced oxidative phosphorylation and favorable mitochondrial membrane potential.f Measurement of glucose uptake by CSps using 2-NBDG ($n = 3$). **g** Lactate release level of CSps ($n = 5$). **h** The ATP levels of CDCs isolated from the CSps of each group ($n = 3$). **i** Representative TEM images of the mitochondrial morphology from each group, and the density of the mitochondria from each group was quantified ($n = 8$ images from 2 experiments). **j** Rhodamine 123 staining results of mitochondrial membrane potential, and the fluorescence intensity of the CDCs isolated from CSps of each group were measured ($n = 15$ images from 3 experiments). All data are shown as the mean \pm SD, * $P < 0.05$ vs. PS, # $P < 0.05$ vs. ULA, one-way ANOVA (f-j) or two-way ANOVA (e).

Methods and Materials

Metabolic analysis of CSps

The glucose uptake ability of CDCs was evaluated by using the fluorescent glucose 2-NBDG (Cayman). After 3 days of cultivation of CDCs on different substrates, all the culture medium was removed and replaced with glucose-free DMEM containing 50 μ M 2-NBDG for 30 min. Consequently, the fluorescence intensity of the cells was measured by flow cytometry (BD FACSCanto). Following a 3-day cultivation, the culture medium of each group was collected to test the extracellular lactate contents, and the reagent kit was the Lactate Colorimetric Assay Kit (Nanjing JianCheng).

For the intracellular ATP content assay, CDCs were cultured on different substrates for 3 days and formed CSps, and single cells were isolated from the CSps of each group and seeded in a 96-well plate at a density of 1×10^5 . The intracellular ATP content was determined by using the CellTiter-Glo Luminescent Cell Viability Assay (Promega). For the mitochondrial membrane potential measurement, digested cells were washed twice with PBS and incubated with 2 μ M Rho123 (Beyotime) for 20 min in a dark environment at 37 $^{\circ}$ C. Fluorescence

images were taken under a fluorescence microscope (Olympus, FV3000), and the fluorescence intensity of the cells was analyzed with ImageJ software.

Oxidative stress resistance level measurement

Following a 3-day cultivation, CDCs formed CSps on different culture substrate, and the CSps were subjected to H₂O₂ stimulation for 24 h. And then singles cells were isolated from the CSps and used for intracellular ROS and superoxide production measurement with the cellular ROS/superoxide detection assay kit (Abcam). In short, the isolated cells were incubated for ROS/superoxide detection for 30 min at 37 °C. The results were observed and recorded by a laser confocal scanning microscope (Olympus, FV3000). In addition, after the CSps were exposed following 12 h or 24 h of exposure to H₂O₂, CSps were digested into single cells and used to measure the cell survival rate with the Alexa Fluor® 488 Annexin-V/Dead Cell Apoptosis Kit (Invitrogen). The results were analyzed by flow cytometry (BD FACSCanto).

7. In the manuscript, cardiac function was measured in post-MI rats by trans-thoracic echocardiography using two-dimensional parasternal long-axis method but the Figure 6 present M-mode results. It should be noted that in ischemic heart failure created by coronary artery ligation, it is not possible to properly evaluate cardiac function using M-mode.

Author Answer: Thank you for your comments, and we are sorry for the unclear expression. There are many different planes could be observed with the two-dimensional ultrasound, and M-mode ultrasound is one of the methods. The purpose of displaying the M-mode ultrasound images in the manuscript was to show the thinning of the anterior wall and decreased motion after myocardial infarction, not to show the ejection fraction of the heart. The size, volume and ejection fraction of the ventricular cavity were calculated using the Simpson method under the four-cavity or two-cavity incision plane of two-dimensional ultrasound, which is used in the present study. The principle of this method is that the volume of a larger geometry can be seen as the volume of several smaller geometers with similar shapes, which could decrease assumptions about geometry and acquire higher accuracy. The procedures of Simpson method are as follows: 1. Obtain four-cavity and two-cavity sections of the apex by ultrasound; 2. Measure Left ventricular end-diastolic volume (LVEDV) and end-systolic volume (LVESV) using the trackball at the 4-chamber and 2-chamber sections of the apex of the heart; 3. Calculate the EF by the equation: $EF\% = (LVEDV - LVESV) / LVEDV * 100\%$. **According to your comment, we have now revised the method section as follows:**

Methods and materials

Echocardiogram

Cardiac function and the movement of the left ventricular wall were measured using a Vevo 2100 ultrasonic system before surgery and at 4, 8, and 12 weeks postsurgery. After the rats were anesthetized with isoflurane, the four-chamber and two-chamber sections of the left ventricle were obtained by ultrasound, left ventricular end-diastolic volume (LVEDV) and end-systolic volume (LVESV) were measured using the trackball at the 4-chamber and 2-chamber sections of the left ventricle, and the left ventricle ejection factor (LVEF) was calculated by the equation: $LVEF = (LVEDV - LVESV) / LVEDV * 100\%$.

8. Measurement of cardiomyocyte size in transversal section is difficult. The most appropriate method is measurement in longitudinal planes combined with immunostaining of intercalated discs.

Author Answer: Thank you for your comments. In fact, there are two major methods to measure cardiomyocyte size, one is to measure the cross-sectional area of cardiomyocytes by WGA staining, and the other is to measure the width of cardiomyocytes by longitudinal planes as you mentioned. To decrease the experimental error, we used the cross section from the same position of the heart in our experiment, and total 100 cells from 8 animals were measured in the experiment. **For your convenience, we have attached the supporting references and related materials of using the cross sections and longitudinal planes for cardiomyocyte size measurement as follows.**

Ref1. Zhang X, et al. Tisp40 prevents cardiac ischemia/reperfusion injury through the hexosamine biosynthetic pathway in male mice. *Nat Commun.* 2023 Jun 8;14(1):3383.

① **Histological analysis:** To evaluate whole morphology, heart samples were excised 4 weeks post-I/R surgery, fixed in 4% neutral formaldehyde for 48 h, dehydrated, embedded in paraffin and sectioned to 5 μm slices. Next, the cardiac slices were incubated with WGA working buffer (1:200) at 37 °C for 1 h to examine the cross-sectional area of cardiomyocytes, and more than 200 cells from 6 mice per group were included.

Fig. g. Heart samples were collected for WGA staining to quantify the cross-sectional area of cardiomyocytes 4 weeks post-I/R surgery (n = 6).

Ref2. Raso A, et al. A microRNA program regulates the balance between cardiomyocyte hyperplasia and hypertrophy and stimulates cardiac regeneration. *Nat Commun.* 2021 Aug 10;12(1):4808.

① **Histological analysis and (immunofluorescence) microscopy:** Hearts were arrested in diastole, perfusion fixed with 4% paraformaldehyde/PBS solution, embedded in paraffin and sectioned at 4 μm . Paraffin sections were stained with FITC-labeled rabbit polyclonal antibody against wheat-germ-agglutinin (WGA) to visualize and quantify the myocyte cross-sectional area (1:100, Sigma Aldrich T4144).

Fig.d wheat germ agglutinin (WGA)-stained (fifth panel) histological sections.

Fig.j quantification of the cell surface areas by wheat germ agglutinin (WGA) staining.

Ref 3. Li S, Nguyen NUN, Xiao F, Menendez-Montes I, Nakada Y, Tan WLW, Anene-Nzulu CG, Foo RS, Thet S, Cardoso AC, Wang P, Elhelaly WM, Lam NT, Pereira AHM, Hill JA, Sadek HA. Mechanism of Eccentric Cardiomyocyte Hypertrophy Secondary to Severe Mitral Regurgitation. *Circulation*. 2020 Jun 2;141(22):1787-1799.

① **Wheat germ agglutinin (WGA) staining and cardiomyocyte size quantification:** WGA staining and quantification was performed as previously described. In brief, the slides were incubated with WGA conjugated to Alexa Fluor 488 (50 mg mL⁻¹, Life Technologies) for 1h at room temperature following washing with PBS. To quantify the cross-sectional cell size, eight independent hearts per group with three different views and positions, each from left ventricle and septum were captured at 40X magnification. ImageJ was used to quantify the size of cardiomyocytes that are in round-shape with the presence of nucleus.

Fig.D WGA staining of sham and MR hearts at 15 weeks after surgery showing similar cardiomyocyte cross-sectional area.

Fig.I Representative immunofluorescence staining confocal images of Sham and MR hearts, stained for Src (red), WGA647(white), a-actinin (green) and DAPI (blue).

9. Is it possible to visualize DiR-labelled cells to explore the phenotype of these grafted cells?

Author Answer: Thank you for your comments. It is a good question. DiR is a lipophilic dye, and it could label the target cell membranes, and be detected under about 700 nm wavelength [Ref1]. It has been reported as a powerful tool for cell tracking. And in our previous study [Ref2], we have used DiR to track the survival of transplanted cells, and trace their cell fates in vivo.

[Ref1] Kalchenko, Vyacheslav; Shivtiel, Shoham; Malina, Victoria; et al. Use of lipophilic near-infrared dye in whole-body optical imaging of hematopoietic cell homing. *Journal of Biomedical Optics*. 11 (2006) 050507.

[Ref2] Zhang, J. et al. Pericardial application as a new route for implanting stem-cell cardiospheres to treat myocardial infarction. *J Physiol* 596, 2037-2054 (2018).

For your convenience, we showed the related reference as follows:

Ref1

A recently developed nearinfrared NIR lipophilic carbocyanine dye 1,1-dioctadecyl-3,3,3,3-tetramethylindotricarbocyanine iodide DiRis used to safely and directly label the membranes of human leukemic Pre-B ALL G2 cell lines as well as primary murine lymphocytes and erythrocytes. DiR has absorption and fluorescence maxima at 750 and 782 nm, respectively, which corresponds to low light absorption and autofluorescence in living tissues. This allows us to obtain a significant signal with very low background level. A charge-coupled device CCD-based imager is used for noninvasive whole-body imaging of DiR-labeled cell homing in intact animals. This powerful technique can potentially visualize any cell type without use of specific antibodies conjugated with NIR fluorescent tag or loading cells with transporter-delivered NIR fluorophores. Thus, in vivo imaging based on NIR lipophilic carbocyanine

dyes in combination with advanced optical techniques may serve as a powerful alternative or complementation to other small animal imaging methods.

Ref 2 in result section

Survival and outcome of CSps after transplantation in vivo

The in vivo survival rate of DiR-labelled CSps at different time points was detected using a small animal in vivo imaging technique. The results are shown in Fig. 4A.

The survival rates of CSps in vivo were $54.5 \pm 7.5\%$ at week 1, $31.0 \pm 4.5\%$ at week 2 and $16.8 \pm 5.3\%$ at week 4. Furthermore, all DiR-labelled CSps were distributed inside cardiac tissue, and no fluorescence was seen in the left or right lung.

The fate of CSps in cardiac tissue after transplantation is shown in Fig. 4B and C. At 4 weeks after transplantation, DiR-labelled cells were clearly seen in cardiac tissue sections, suggesting survival in cardiac tissue in vivo. Specifically, these cells reached the epicardium (Fig. 4B), and also migrated into the infarcted myocardial tissues (Fig. 4C). These data show that the implanted CSps were maintained well in the heart with no loss to other tissues and were able to migrate to myocardial tissue and infiltrate into the infarcted area.

Figure 4. Survival and the outcome of CSps after transplantation in vivo. A, survival rate of DiR-labelled CSps in vivo at week 1, 2 and 4; B, DiR-labelled CSps survived and infiltrated to the epicardium at week 4; C, DiR-labelled CSps migrated into infarcted myocardial tissue at week 4. Immunofluorescence data: DiR=1,1-dioctadecyl-3,3,3-tetramethylindotricarbocyanine iodide, DAPI= 4,6-diamidino-2-phenylindole, cTnT = cardiac troponin T. Note, DAPI and cTnT images show loss of myocardial cells in the infarcted myocardial tissue.

Response to reviewer 3:

1. In the article, the author stated that the peripheral cells of OPC-CSps in the PS group, ULA group and OPC group showed pyroptosis positive. Has the author tested and compared the expression of other types of death methods such as necroptosis, apoptosis or ferroptosis? Why do research on pyroptosis directly.

Author Answer: Thank you for your comment. It is an interesting question. Apoptosis, pyroptosis, necroptosis and ferroptosis are programmed cell death pathways, and they are greatly related to inflammation. In our preliminary experiment, we observed partial cell death following cells seeding on the OPC culture, and we tested several inflammatory factors and significant upregulation of IL-1beta was observed. It is well known that IL-1beta is a typical regulator of pyroptosis, so we further investigated the pyroptosis pathway in OPC-CSps in the present study. Still, there are possibilities that other cell death pathways also involved in the regulation of OPC-CSps, and we will continuously study these problems in our future study according to your comment.

2. The KEGG results in Figure 4c show that the HIF-1 signaling pathway is enriched, whether to verify the activation of this pathway in vivo and in vitro.

Author Answer: Thank you for your comment. HIF-1 signaling pathway is important in the response to low oxygen levels or hypoxia. In the present study, CSps cultured on the OPC substrate were observed to form 70–140 μm in diameter, and the internal cells in the OPC-CSps maintained normal cell ultrastructure. In contrast, the ULA spheroids were 100-470 μm in diameter which is beyond the size of effective nutrient and oxygen transportation. In addition, the GSEA analysis result revealed the significant upregulation of the hypoxic components in the ULA-CSps when compared to the OPC group and the PS group, respectively. And no significant difference was observed between the OPC group and the PS group (Fig. 4d, $P=0.06$), so the hypoxia level could be briefly concluded as $\text{ULA} > \text{OPC} = \text{PS}$. It is a good idea to verify the activation of HIF-1 signaling pathway in the CSps, and we will design the in vivo and in vitro experiments based on your opinion in the future studies.

3. The author stated that oxidative stress injury and inflammation play a key role in acute MI, but in vivo experiments only verified the ability of OPC to resist oxidative stress injury, and its anti-inflammatory effect in vitro did not appear to be superficial.

Author Answer: Thank you for your comment. It is a good idea to verify the anti-inflammatory effect in vitro. According to your suggestion, we have replenished the

corresponding examination in this study. TNF-alpha was used to stimulate the cells isolated from different groups for 12 h, and cell viability was analysis by annexin V/PI. The results showed that compared to the control group, all the groups committed to TNF-alpha treatment showed significant decrease of cell viability, while the cells from the OPC-CSps group exhibited the best anti-inflammatory effect among 3 groups. **The corresponding revisions are as follows:**

Results

OPC-CSps showed enhanced oxidative stress resistance and anti-inflammatory effect

... The anti-inflammatory effects were also examined. Following 12 h of TNF- α stimulation, the survival rates of the PS, ULA, and OPC groups were $50.0\% \pm 3.7\%$, $53.8\% \pm 4.5\%$, and $66.3\% \pm 3.0\%$, which were significantly lower than the control group, while the cells from the OPC-CSps group exhibited the best anti-inflammatory effect among 3 groups (Fig. 5e).

Figure 5. Improvement in oxidative stress resistance in OPC-CSps following H₂O₂-induced injury. ... (e) The percentages of viable cells from each group following 12 h of TNF- α stimulation were determined by Annexin V/PI flow cytometry analysis ($n = 3$). All data are shown as the mean \pm SD, * $P < 0.05$ vs. Control, # $P < 0.05$ vs. PS, & $P < 0.05$ vs. ULA, one-way ANOVA.

Methods and materials

Anti-inflammation ability measurement

Following a 3-day cultivation, CDCs formed CSps on different culture substrate, and the CSps were subjected to 50 ng/ml TNF- α stimulation for 12 h. And then singles cells were isolated from the CSps and used to measure the cell survival rate with the Alexa Fluor® 488 Annexin-V/Dead Cell Apoptosis Kit (Invitrogen). The results were analyzed by flow cytometry (BD FACSCanto).

4. In Figure 7b, the author only uses the amount of CD68 to represent inflammation, whether it can represent the polarization of macrophages, and to improve the effect of OPC-CSps on the polarization of macrophages in vitro.

Author Answer: Thank you for your question, this issue is very important. Because the macrophage phenotype is important for the regulation of cardiac regenerative repair. CD68 is one of the markers of macrophages, and Geoffrey de Couto et al. demonstrated that CDCs reduced the number of CD68+ macrophages in the infarcted

myocardium and polarized macrophages away from the M1 phenotype but not towards the classical M2 state. [Ref 1] Instead, they polarized to a distinct cardioprotective phenotype that promotes the survival of ischemic cardiomyocytes. We also expand the discussion on the role of CDCs on macrophage polarization in the article. and your suggestion is very enlightening, and we will continue to explore this issue in our subsequent studies. **The revisions are as follows:**

Discussion, paragraph 9

...In this study, owing to the effective preconditioning treatment in vitro, the OPC-CSps acquired enhanced inflammation resistance and improved regenerative function. The transplanted CSps were tracked by Dir labeling, and the results showed that $16.18\% \pm 3.68\%$ of CSps survived 4 weeks following transplantation (Fig. 6a&6b). Meanwhile, a significant decrease in CD68⁺ macrophages in the border zone was observed in the OPC-CSps group (Fig. 7b&7f). Interestingly, Geoffrey de Couto et al. also demonstrated reduction of the CD68⁺ macrophages in the ischemic heart following CDCs transplantation, and they proposed that the CDCs polarized macrophages away from the M1 phenotype but not toward a classical M2 state, but to a distinct cardioprotective phenotype that promotes the survival of ischemic cardiomyocytes⁵¹. Moreover, effective angiogenesis within the infarcted myocardium was widely observed at 12 weeks following MI (Fig. 7b&7g). Therefore, these in vivo results demonstrated that transplanting OPC-CSps could achieve satisfactory long-term cardiac function recovery by effectively reducing myocardial inflammation and promoting angiogenesis.

51. de Couto G. *et al.* Macrophages mediate cardioprotective cellular postconditioning in acute myocardial infarction. *J Clin Invest* **125**, 3147-3162 (2015).

Reference in the response

[Ref 1] de Couto G, Liu W, Tseliou E, Sun B, Makkar N, Kanazawa H, Arditi M, Marbán E. Macrophages mediate cardioprotective cellular postconditioning in acute myocardial infarction. *J Clin Invest*. 2015 Aug 3;125(8):3147-62.

For your convenience, we showed the related reference as follows:

Results:

CDCs reduce the number of CD68⁺ macrophages within the ischemic heart.

To test the idea that CDCs modulate inflammation following IR, we examined the leukocyte profile of peripheral blood and cardiac tissue (Figure 4A). Delivery of CDCs to the heart altered neither circulating leukocytes (Supplemental Table 2) nor serum expression of proinflammatory MCP-1 or IL-4 (Figure 4E). It did, however, reduce specific leukocyte populations within the heart, notably CD45⁺CD68⁺ macrophages (Figure 4B) and CD45⁺CD11b⁺CD11c⁺ dendritic cells (Supplemental Table 3); both are members of the mononuclear phagocyte system. Interestingly, granulocytes (CD45⁺Gran), another acute infiltrating inflammatory cell type (2), were unaltered (Figure 4B). These data were validated using immunohistochemistry to detect

CD68⁺ within the infarct region of hearts isolated 2, 6, and 48 hours following IR (Figure 4, C and D). While the CD68⁺ cell number was similar in hearts treated with CDC or PBS at 2 and 6 hours after IR, the number of CD68 cells (macrophages) was reduced at 48 hours in the hearts of CDC-treated animals (Figure 4, C and D). Furthermore, we did not observe changes in circulating levels of MCP-1 (CCL2) or IL-4 (Figure 4E), which might have accounted for the altered macrophage influx between treatment groups.

Figure 4 CDC-treated animals have a reduced CD68⁺ macrophage population 48 hours after IR.

(A) Gating strategy for leukocyte identification within the infarcted myocardium prior to subset analysis. CD45 cells were first identified (FSC-A/CD45), and then dead cells were excluded (DAPI⁺⁺⁻). (B) Pooled flow cytometry data from infarcted rat tissue reveal a reduced CD68 population in CDC- vs. PBS-treated hearts (n = 4 rats per group). (C) Immunohistochemical staining of hearts within the infarct zone from CDC- and PBS-treated animals at 2, 6, and 48 hours after IR (n = 4 rats per group). Scale bar: 50 μ m. (D) Pooled data from CD68 cells within the infarct tissue from C at 2, 6, and 48 hours after IR (n = 4 rats per group). (E) Concentration of cytokines MCP-1 (CCL2) and IL-4 in sera 48 hours after IR (n = 4–8 rats per group). Graphs depict mean \pm SEM. Statistical significance was determined using Student's t test and 2-way ANOVA followed by Bonferroni's multiple comparisons test. *P < 0.05.

Result:

CDCs shift the cardiac CD68⁺ macrophage population away from an M1 phenotype in vivo.

It is well recognized that macrophages exhibit the capacity to polarize between M1 and M2 phenotypes (6, 24). The M1 population is generally defined by its early infiltration into the myocardium and proinflammatory cytokine expression (e.g., Nos2, Tnf, Il1b, and Il6), while the M2 population is associated with resolution of late-phase inflammation and promotion of tissue repair (e.g., Arg1, Il10, and Pparg). We therefore asked whether CDCs polarize macrophages toward the M1 or M2 phenotype. To do so, we induced MI by permanently ligating the left anterior coronary artery and randomly allocated rats to receive 2×10^6 CDCs or an equivalent volume of vehicle (PBS) through direct injection at 4 sites in the border zone immediately following infarction. Two days later, hearts were harvested and the infarct and surrounding border

tissue were digested. The resulting cell suspension was separated using a density gradient to isolate the mononuclear cell fraction, and then cardiac macrophages were purified by attachment on cell culture dishes (Figure 6A). The >85% pure CD68 populations were then analyzed by qRT-PCR for M1 and M2 gene expression markers (+Figure 6B). Interestingly, M1 markers, Nos2, Tnf, and Il1b, were significantly reduced, but there was no concomitant increase in M2 markers, such as Arg1, Il10, Il4ra, or Pparg. Thus, CDCs reduce the number of CD68+ macrophages in the infarcted myocardium and polarize macrophages away from the M1 phenotype but not toward a classical M2 state.

Figure 6 Cardiac macrophages isolated from CDC-treated animals have a distinct cytokine profile. (A) Representative images and pooled quantitative analysis of CD68 macrophages isolated from cardiac tissue of PBS- and CDC-treated animals 48 hours following MI. Immunohistochemistry reveals a purity level of >85% CD68 positivity following cardiac macrophage isolation (n = 3 rats per group). Scale bar: 10 μ m. (B) Gene expression profile from cardiac macrophages isolated from infarcted hearts after 48 hours. CDC-treated hearts have cardiac macrophages with reduced M1 (Tnf, Nos2, Il1a, and Il1b), but no change in M2 (Arg1, Tgfb1, Il10, and Pparg), macrophage gene expression. Graphs depict mean \pm SEM. Statistical significance was determined using 2-way ANOVA followed by Bonferroni's or Sidak's multiple comparisons test. *P < 0.05. M ϕ , macrophage.

5. In Figure 7b, the author only shows blood vessels by single staining of α -SMA. Is it not rigorous enough? It is recommended to co-stain with endothelial cell indicator CD31, and double positives indicate blood vessels.

Author Answer: Thank you for your comments. α -SMA is a marker of It is a good idea to co-stain CD31 and α -SMA to indicate the blood vessels in the infarcted myocardium, and the corresponding revisions are as follows:

Results

OPC-CSps protected infarcted myocardium from inflammation, apoptosis, and hypertrophy

...The structure and distribution of the vessels in the LV wall were observed by α -SMA and CD31 immunofluorescence co-staining. The vessel lumen could be obviously observed in the sham, vehicle, and OPC-CSps groups. Compared to the sham group, the vessel density of the vehicle group and the OPC-CSps group both significantly increased ($P<0.05$). Compared to the vehicle group, a significantly higher vascular density ($P<0.05$) and mature large-diameter blood vessels ($>100\ \mu\text{m}$) were observed in the OPC-CSps groups (Fig. 7b&7g, Supplementary Fig. 5).

Figure 7. The reduction in cardiac inflammation, apoptosis, and hypertrophy after OPC-CSps transplantation. ...b Representative images of CD68⁺ macrophages in the peri-infarct zone 2 weeks following MI, α -SMA⁺ CD31⁺ vessels in the infarct zone 12 weeks following MI, and TUNEL⁺ cells in the peri-infarct zone 12 weeks following MI. Spilt channels of the α -SMA and CD31 staining results are shown in Supplementary Fig. 5.

Supplementary Fig. 5. Spilt channels images of the α -SMA and CD31 immunofluorescence staining results of Fig. 7b. Paraffin sections of the hearts from each group were used for analysis. The cell nucleus was detected by staining with Dapi (blue), and the expression of anti-CD31 (endothelial cell marker) antibodies (green) and anti- α -SMA (smooth muscle cell marker) antibodies (red) were co-stain to determine the vascular distribution of the left ventricle area.

Methods and materials

Histology and immunochemistry assay

For cell sample preparation, CSps from each group were harvested, fixed in a 75% ethanol solution, and embedded in OCT compound, and 5 μ m sections were used. For tissue sample preparation, after rats were euthanized, the heart tissues were harvested, fixed in 4% paraformaldehyde, embedded in paraffin, and serially sectioned at 5 μ m thickness. Masson trichrome staining was performed on tissue sections. For immunofluorescence staining, the sections were blocked with 5% goat serum and then incubated with primary antibodies, including anti-caspase1(1:100, ab1872, Abcam), anti-CD68 (1:200, 360018, Zhengneng), anti-cardiac troponin T (cTnT, 1:500, ab209813, Abcam), anti-smooth muscle alpha-actin (α -SMA, 1:300, 41550, SAB), and anti-CD31 (1:300, GB11063-3, Servicebio) at 4 $^{\circ}$ C overnight. After washing with PBS 3 times, 488 nm goat anti-rabbit (1:400, ab150077, Abcam), 594 nm goat anti-rabbit (1:400, ab150080, Abcam), 488 nm goat anti-Mouse(1:400, ab150113, Abcam), or 594 nm goat anti-Mouse (1:400, ab150116, Abcam) secondary antibodies were added and incubated for 2 h at room temperature.....

6. The author stated in the previous article that OPC has the activation of pyroptosis, whether the expression level of pyroptosis is expressed in the subsequent in vivo and in vitro verification, and whether OPC-CSps has an impact on pyroptosis.

Author Answer: Thank you for your comments. It is a good question. Firstly, following culture on the OPC substrate, only the periphery cells in the CSps are induced to pyroptosis. And it is important to strengthen that the purpose of preconditioning treatment was to provide proper inflammatory stimulation for the majority cells in the CSps to enhance their cellular function and anti-oxidative resistance.

To evaluate the entire cellular function of CSps in this study, the CDCs were isolated from the CSps of each group and used for the examinations of proliferation assay and mitochondrial function test. In addition, the oxidative stress resistance examination and anti-inflammatory effect assay were performed with the CSps, and single cells were digested from the CSps following stimulation and used for the following flow cytometry analysis. The results showed improved proliferation, mitochondria function, anti-oxidative resistance, and anti-inflammatory effect of the cells in OPC-CSps compared to the PS group and the ULA group. To conclude, there may be few pyroptotic cells in the OPC-CSps, but the entire cellular functions were enhanced.

Secondly, when MI occurs, myocardial ischemia causes a series of irreversible pathological processes, such as severe inflammation and massive cell death. The severe inflammatory microenvironment in the infarcted area is a major challenge for transplanted cell survival, and the in vivo inflammation level of the is much greater than the partial pyroptotic cell of OPC-CSps. And the OPC-CSps also secreted a variety of growth factors that are beneficial for cardiac regeneration. Taken together, the OPC-CSps would not have impact on inducing pyroptosis in vivo, but play a role in improving cardiac function.

7. The author should add PS and ULA groups for comparison in the verification of in vivo, not just in vitro.

Author Answer: Thank you for your comments. It is reasonable to compare the CSps prepared by different approaches and observed their therapeutic effects on MI. However, in accordance to the 3R principle of animal experiments (reduction, replacement and refinement), the present study verified the functions of CSps (CDCs) among different groups in vitro before performing the in vivo studies. The in vitro studies evidenced that CDCs from the OPC-CSps showed improvements of cell survival, proliferation ability, and growth factors paracrine levels. Also, improved mitochondrial function and enhancement of oxidative stress resistance of the CDCs from OPC-CSps were observed. Taken the in vitro experiments results together, OPC-CSps showed superior bioactivities compared to the PS-CSps and ULA-CSps, so the OPC-CSps was further examined its therapeutic effects on MI rats.

REVIEWERS' COMMENTS

Reviewer #1 (Remarks to the Author):

Thankyou for your detailed discussion of my comments, I think the manuscript is much improved

Reviewer #2 (Remarks to the Author):

- I disagree with the authors regarding the answer to comment 8. The most appropriate method for measuring cardiomyocyte size is the measurement of cardiomyocyte size in longitudinal planes combined with immunostaining of intercalated discs. With this method, we are able to measure length and width of cardiomyocyte as well as the length-to-width ratio. This comment should be taken into account.

- I still believe it will be useful to explore the phenotype of the grafted DIR+ cells..... particularly to evaluate the impact of OPC.

- Please correct the expression "before MI" in figure 6. With this terminology, we understand that Sham also undergoes myocardial infarction.

Reviewer #3 (Remarks to the Author):

The authors have appropriately addressed all my comments and I have no further concerns.

RESPONSES TO REVIEWERS' COMMENTS

We appreciate your prompt reviews and valuable suggestions, and we had revised the manuscript according to your comments and suggestions. All revised text was highlighted in yellow color in the revised manuscript, and the grammar and spell error were showed in red color.

Response to reviewer 2:

1. I still believe it will be useful to explore the phenotype of the grafted DiR+ cells..... particularly to evaluate the impact of OPC.

Author Answer: Thank you for your comments. In our previous study, by using DiR labelling in combination with immunofluorescence, the transplanted DiR-labelled CSps showed cTNT-positive, indicating the direct differentiation of CSps into cardiomyocytes. Since we transplanted the same cell types in two studies, so the cell fate would be the same. Still, it is a good idea to compare the differentiation abilities of PS-CSps and OPC-CSps into cardiomyocytes in vivo, and we are going to study this problem in the future. To better illustrate this problem, we have replenished the study bases of the grafted DiR-labelled CSps in vivo. Combining the editorial's office suggestion, we have summarized the previous studies of cardiospheres in the Introduction section, **the revised version is as follows:**

Cardiosphere-derived cells (CDCs) are of endogenous cardiac origin²⁰ and possess the ability to form 3D spherical clones, cardiospheres (CSps), in vitro²¹. Compared to monolayer CDCs, CSps possessed improved growth factor secretion and cardiac regeneration potential, making them a good cell source for MI therapy^{22, 23}. Previous reports demonstrated that CSps could regulate the inflammation of infarcted myocardium through immunomodulatory effects^{24, 25, 26}, and it was proposed that the CDCs polarized macrophages away from the M1 phenotype but not toward a classical M2 state, but to a distinct cardioprotective phenotype that promotes the survival of ischemic cardiomyocytes²⁷. Also, the secretion of various growth factors and bioactive molecules from CSps could involve in the vessel network rebuilding process, which were beneficial for reducing ventricular adverse remodeling and hypertrophy^{28, 29, 30, 31}. These characteristics make CSps a good cell source for MI therapy. Previously, preconditioning CSps with pericardial fluid obtained from myocardial infarction was prepared by our colleagues Zhang et al., and the paracrine function and survival rate of the pericardial fluid-pretreated CSps dramatically increased, exhibiting significant improvement of MI cardiac function, and the DiR-labelled CSps showed cTnT-positive in the infarcted area, indicating the direct differentiation of CSps into cardiomyocytes³². Moreover, Zhang et al. reported that pericardial application could serve as a new and effective route for CSps

transplantation, and this therapeutic strategy also showed favorable potential for further clinical application³³.

2. Please correct the expression "before MI" in figure 6. With this terminology, we understand that Sham also undergoes myocardial infarction.

Author Answer: Thank you for your careful review. We appreciate your suggestion, and we have now revised the expression "before MI" to "before surgery" in Figure 6. **The revised version is as follows:**